# DATASET FAIRNESS: ACHIEVABLE FAIRNESS ON YOUR DATA WITH UTILITY GUARANTEES

## ABSTRACT

In machine learning fairness, training models which minimize disparity across different sensitive groups often leads to diminished accuracy, a phenomenon known as the fairness-accuracy trade-off. The severity of this trade-off fundamentally depends on dataset characteristics such as dataset imbalances or biases, and therefore using a universal fairness requirement across datasets remains questionable and can often lead to models with varying and substantially low utility. To address this, we present a computationally efficient approach to approximate the fairness-accuracy trade-off curve tailored to individual datasets, backed by rigorous statistical guarantees. By utilizing the You-Only-Train-Once (YOTO) framework, our approach mitigates the computational burden of having to train multiple models when approximating the trade-off curve. Moreover, we introduce confidence intervals around this curve, offering a statistically grounded perspective on acceptable range of fairness violations for any given accuracy threshold. Our empirical evaluation which includes applications to tabular data, computer vision and natural language datasets, underscores that our approach can guide practitioners in accuracy-constrained fairness decisions across various data modalities.

## 1 INTRODUCTION

One of the key challenges in fairness for machine learning is to train models that minimize the disparity across various sensitive groups such as race or gender (Caton & Haas, 2020; Ustun et al., 2019; Celis et al., 2019). This often comes at cost of reduced model accuracy, a phenomenon termed fairness-accuracy trade-off in literature (Valdivia et al., 2021; Martinez et al., 2020). In practice, this trade-off can differ significantly across datasets, depending on factors such as dataset biases, imbalances etc. (Agarwal et al., 2018; Bendekgey & Sudderth, 2021; Celis et al., 2021).

This raises significant challenges for deploying these models in practical settings. For instance, it is not evident whether one should adopt the same disparity threshold for different tasks. Consider two crime datasets: Dataset A has records from a community where crime rates are uniformly distributed across all racial groups, whereas dataset B comes from a community where historical factors have resulted in a disproportionate crime rate among a specific racial group. Intuitively, training models which are racially agnostic is more challenging for the latter, due to the unequal distribution of crime rates across racial groups in the former. Thus, applying uniform fairness guidelines to both datasets necessitates careful consideration to account for their distinct characteristics and underlying biases.

This example underscores one of the main challenges in fairness for machine learning models. More specifically, setting a uniform requirement for fairness while also adhering to essential accuracy benchmarks is impractical across diverse datasets. Hence, the literature requires a principled guideline for the range of achievable fairness violations. To put it concretely, the question becomes:

*For a given dataset, model class, and accuracy, what is the range of permissible fairness violation?*

One way to answer this question is by having access to the ground truth fairness-accuracy trade-off curve (see Fig 1), where the curve shows for each attainable accuracy, what the minimum achievable fairness violation is, i.e. this could serve as a manual to look up reasonable fairness violations given a target accuracy. Unfortunately this curve is often unavailable and hence, various optimization techniques have been proposed to approximate the curve ranging from regularization (Bendekgey & Sudderth, 2021; Olfat & Mintz, 2020) to adversarial learning (Zhang et al., 2018; Yang et al., 2023).

Nevertheless, the problem with these aforementioned methods is that recovering the whole curve can be computationally very expensive as it essentially requires retraining hundreds if not thousands of models to obtain a good approximation of the trade-off curve (see Figure 1 where each dot corresponds to a separately trained model).

In this paper, we introduce a computationally efficient method to approximate the optimal fairness-accuracy trade-off curve, supported by rigorous statistical guarantees. On a high level, our method is divided into two steps. Firstly, our approach adapts a technique from Dosovitskiy & Djolonga (2020) called You-Only-Train-Once (YOTO) to the fairness setting. This framework allows us to train only one model that can represent a range of fairness-accuracy trade-offs, thereby significantly reducing computational demands. Details on the YOTO framework and how it is incorporated in our setting will be discussed in Section 3.1. Secondly, once we obtain an estimate of the trade-off curve, we turn to its reliability. Inspired by recent work on risk-controlling prediction sets (Bates et al., 2021), we construct confidence intervals using the estimated trade-off curve and theoretically prove that the ground truth curve will lie in the interval with high probability. This gives us statistically backed evidence to express the range of admissible fairness violations conditioned on a given accuracy, allowing us to make statements such as:

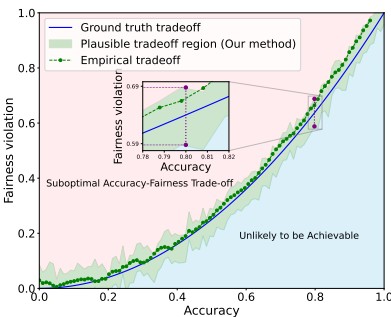

Figure 1: The ground truth and estimated trade-offs between accuracy and fairness. The green area depicts the range of admissible fairness violations for a given accuracy. The pink area shows suboptimal accuracy-fairness trade-offs, while the blue area shows unlikely-to-be-achieved ones.

*For a given dataset, model class, and accuracy, the permissible range of fairness violation is x to y.*

Hence, our proposed intervals together with the estimated trade-off curve, allow practitioners to decide whether the current model meets acceptable standards of fairness given the dataset, model class, and accuracy constraints. The contributions of this paper are three-fold:

- We present a methodology of obtaining a range of permissible fairness violations for any given dataset and any desired model accuracy chosen dynamically at inference time, which provides practitioners with a convenient tool for fairness decision-making.
- To do so, we introduce a technical framework to construct confidence intervals with statistical guarantees on the optimal fairness-accuracy trade-off curve in a computationally efficient way. Moreover, we extend our intervals to the setting where sensitive attributes are scarce in the data.
- Lastly, we empirically show across various data modalities that our intervals indeed contain the trade-off curves of SOTA fairness methods, ranging from regularized to adversarial methods.

**Outline**: Section 2 introduces the notation and problem setup. Section 3 details our proposed method for establishing computationally efficient and statistically valid confidence intervals on the true trade-off curve. Section 4 discusses related work and Section 5 provides empirical evaluation of our method. The paper concludes in Section 6, where we discuss limitations and future work.

## 2 PRELIMINARIES

**Notation** Throughout this paper, we consider a binary classification, where each training sample is composed of triples, $(X, A, Y)$. $X \in \mathcal{X}$ denotes a vector of features, $A \in \mathcal{A}$ indicates a sensitive attribute, and $Y \in \mathcal{Y} := \{0, 1\}$ represents a label. To make this more concrete, if we take loan default prediction as the classification task, an individual could be represented such that their income level, loan amount, and previous payment records are embodied in $X$; their racial identity is represented by $A$; and their loan default status is $Y$. Having established the notation, for completeness, we provide some commonly used fairness violations $\Phi_{\text{fair}}(h) \in [0, 1]$ in the setting when $\mathcal{Y} = \mathcal{A} = \{0, 1\}$:

**Demographic Parity (DP):** DP condition states that the selection rates for all sensitive groups are equal, i.e. $\mathbb{P}(h(X) = 1 \mid A = a) = \mathbb{P}(h(X) = 1)$ for any $a \in \mathcal{A}$. The absolute DP violation is:

$$\Phi_{\text{DP}}(h) := |\mathbb{P}(h(X) = 1 \mid A = 1) - \mathbb{P}(h(X) = 1 \mid A = 0)|.$$

**Equalized Opportunity (EOP):**   EOP condition states that the true positive rates for all sensitive groups are equal, i.e. $\mathbb{P}(h(X) = 1 \mid A = a, Y = 1) = \mathbb{P}(h(X) = 1 \mid Y = 1)$ for any $a \in \mathcal{A}$. The absolute EOP violation for the case when $\mathcal{A} = \{0, 1\}$ is defined as:

$$\Phi_{\text{EOP}}(h) \coloneqq |\mathbb{P}(h(X) = 1 \mid A = 1, Y = 1) - \mathbb{P}(h(X) = 1 \mid A = 0, Y = 1)|.$$

## 2.1   PROBLEM SETUP

For a model class $\mathcal{H}$ (e.g., neural networks) and a given accuracy threshold $\psi \in [0, 1]$, we define the optimal accuracy-fairness trade-off $\tau_{\text{fair}}^*(\psi)$ as the minimum attainable fairness violation, $\min_{h \in \mathcal{H}} \Phi_{\text{fair}}(h)$, subject to the constraint that the accuracy of the model is above a certain threshold $\psi$, i.e. $\text{acc}(h) \geq \psi$, where $\text{acc}(h)$ is the accuracy of a model $h$, i.e.

$$\tau_{\text{fair}}^*(\psi) \coloneqq \min_{h \in \mathcal{H}} \Phi_{\text{fair}}(h) \quad \text{subject to} \quad \text{acc}(h) \geq \psi. \tag{1}$$

For an unattainable accuracy threshold $\psi'$, we define $\tau_{\text{fair}}^*(\psi') = 1$. Given any dataset, the trade-off curve $\tau_{\text{fair}}^*$ helps us characterize *dataset fairness*, a notion we use to describe the fairness properties of the dataset. Our goal is to reliably estimate this trade-off curve $\tau_{\text{fair}}^* : [0, 1] \to [0, 1]$. This is contrary to Agarwal et al. (2018), which optimize model accuracy subject to fairness constraints.

Obtaining the exact ground-truth trade-off curve $\tau_{\text{fair}}^*$ defined in Eq. (1) is inherently challenging for several reasons. First, we are limited by the confines of a finite dataset, which restricts our ability to compute the exact values of accuracy $\text{acc}(h)$ and fairness violation $\Phi_{\text{fair}}(h)$. Second, the constrained optimization problem in Eq. (1) required to obtain the value of $\tau_{\text{fair}}^*(\psi)$ is non-trivial to solve exactly and may require training separate models across different accuracy constraints.

**High-level road map:** Given these limitations, we seek to quantify the plausible range of optimal accuracy-fairness trade-offs for a given dataset using a two-step approach:

1. Firstly, we empirically estimate the trade-off curve $\tau_{\text{fair}}^*$ using YOTO (Dosovitskiy & Djolonga, 2020), a computationally efficient methodology that avoids having to train multiple models.

2. Secondly, we obtain valid confidence intervals on the minimum attainable fairness violation $\tau_{\text{fair}}^*(\psi)$ using a held-out calibration dataset, denoted as $\mathcal{D}_{\text{cal}}$. Specifically, given $\alpha \in (0, 1)$, we construct confidence intervals $\Gamma_{\text{fair}}^\alpha \subseteq [0, 1]$ which satisfies guarantees of the form:

$$\mathbb{P}(\tau_{\text{fair}}^*(\Psi) \in \Gamma_{\text{fair}}^\alpha) \geq 1 - \alpha.$$

   Here, $\Gamma_{\text{fair}}^\alpha$ is computed using a held-out calibration dataset $\mathcal{D}_{\text{cal}}$, $\Psi \in [0, 1]$ are random variables obtained using calibration data $\mathcal{D}_{\text{cal}}$. The probability guarantee above is marginal over both $\Psi$ and $\Gamma_{\text{fair}}$, which is analogous to the guarantees obtained using conformal prediction (Vovk et al., 2005; Angelopoulos & Bates, 2021). For more details see Section 3.2.

## 3   METHODOLOGY

In this section, we will demonstrate how the above mentioned 2-step approach offers a practical as well as statistically sound methodology for reliable estimation of $\tau_{\text{fair}}^*(\psi)$. Figure 1 provides an illustration of our proposed confidence intervals $\Gamma_{\text{fair}}^\alpha$ and shows how they can be interpreted as a range of 'admissible' values of fairness violations for models $h$ with $\text{acc}(h) \geq \psi$. In particular, if for a classifier $h_0$ with $\text{acc}(h_0) \geq \psi$, the fairness violation value $\Phi_{\text{fair}}(h_0)$ lies above the confidence intervals $\Gamma_{\text{fair}}^\alpha$ (i.e., the pink region in Fig. 1), then $h_0$ is likely to be suboptimal in terms of the fairness violation, i.e., $\Phi_{\text{fair}}(h_0)$ can be reduced while keeping the accuracy fixed. On the other hand, the fairness violation below the confidence interval $\Gamma_{\text{fair}}^\alpha$ (the blue region in Figure 1) is unlikely to be achieved by models with $\text{acc}(h) \geq \psi$. Next, we outline how to construct such intervals.

## 3.1   STEP1: EFFICIENT ESTIMATION OF TRADE-OFF CURVE

The first step of constructing the intervals is to approximate the trade-off curve by recasting the problem into a constrained optimization objective. The optimization problem formulated in Eq. (1) is however, often too complex to solve, because the accuracy $\text{acc}(h)$ and fairness violations $\Phi_{\text{fair}}(h)$ are both non-smooth (Agarwal et al., 2018; Bendekgey & Sudderth, 2021). These constraints make

it hard to use standard optimization methods that rely on gradients (Kingma & Ba, 2014). To get around this issue, previous works in the fairness literature (Agarwal et al., 2018; Bendekgey & Sudderth, 2021) replace the non-smooth constrained optimisation problem with a smooth surrogate loss. Here, we consider parameterized family of classifiers $\mathcal{H} = \{h_\theta : \mathcal{X} \to \mathbb{R} \mid \theta \in \Theta\}$ (such as neural networks) with predictions $\hat{Y} = \mathbb{1}(h_\theta(X) > 0)$, trained using the regularized loss:

$$\mathcal{L}_\lambda(\theta) = \mathbb{E}[\mathcal{L}_{\text{CE}}(h_\theta(X), Y)] + \lambda \, \mathcal{L}_{\text{fair}}(h_\theta). \tag{2}$$

where, $\mathcal{L}_{\text{CE}}$ is the cross-entropy loss for the classifier $h_\theta$ and $\mathcal{L}_{\text{fair}}(h_\theta)$ is a smooth relaxation of the fairness violation $\Phi_{\text{fair}}$ (Bendekgey & Sudderth, 2021; Lohaus et al., 2020). The parameter $\lambda \in \mathbb{R}_{\geq 0}$ in $\mathcal{L}_\lambda$ modulates the accuracy-fairness trade-off with lower values of $\lambda$ favouring higher accuracy over reduced fairness violation and vice-versa. Therefore, given an (achievable) accuracy threshold $\psi$, there exists a value of $\lambda \geq 0$ such that the loss $\mathcal{L}_\lambda(\theta)$ in Eq. (2) provides a smooth surrogate loss for the constrained optimisation problem in Eq. (1). For more details, we refer the interested reader to Bendekgey & Sudderth (2021) for examples of such regularizers.

Now that we defined the optimization objective, obtaining the trade-off curve becomes straightforward by simply optimizing over a grid of $\lambda$'s. Let $\theta_\lambda^* \in \Theta$ be the minimiser of loss $\mathcal{L}_\lambda$, i.e. $\theta_\lambda^* = \arg\min_{\theta \in \Theta} \mathbb{E}[\mathcal{L}_{\text{CE}}(h_\theta(X), Y)] + \lambda \, \mathcal{L}_{\text{fair}}(h_\theta)$, then we can approximate $\tau_{\text{fair}}^*(\psi)$ pointwise by:

$$\tau_{\text{fair}}^*(\psi) \approx \min_{i \in \{1, \dots, k\}} \left\{ \widehat{\Phi_{\text{fair}}}(h_{\theta_{\lambda_i}^*}) \mid \widehat{\text{acc}}(h_{\theta_{\lambda_i}^*}) \geq \psi \right\}, \tag{3}$$

where $\widehat{\Phi_{\text{fair}}}(h_\theta)$ and $\widehat{\text{acc}}(h_\theta)$ denote the empirical fairness violation and accuracy of the classifier $\hat{Y} = \mathbb{1}(h_\theta(X) > 0)$ on some held-out data. However, training multiple models can be computationally expensive, especially when the model class $\mathcal{H}$ are large-scale models (e.g. neural networks). Moreover, the accuracy and fairness violations $\text{acc}(h_{\theta_\lambda^*}), \Phi_{\text{fair}}(h_{\theta^*})$ may not vary continuously with changing values of $\lambda$, and a small increase in the value of $\lambda$ could lead to a large shift in the accuracy and fairness violations (Bendekgey & Sudderth, 2021). Consequently, it is challenging to find models which have an accuracy close to a given value $\psi$ as the $\lambda$ parameter offers little control over the model accuracy $\text{acc}(h_{\theta_\lambda^*})$. Next, to circumvent these challenges, we employ neural networks with loss-conditional training which were originally proposed by Dosovitskiy & Djolonga (2020).

### 3.1.1 Loss-conditional fairness training

As we describe above, a popular approach for approximating the accuracy-fairness trade-off $\tau_{\text{fair}}^*(\psi)$ involves training multiple models $h_{\theta_\lambda^*}$ over a discrete grid of $\lambda$ hyperparameters with the regularized loss $\mathcal{L}_\lambda$. To avoid the computational overhead of training multiple models, Dosovitskiy & Djolonga (2020) propose 'You Only Train Once' (YOTO) a methodology of training one model $h_\theta : \mathcal{X} \times \Lambda \to \mathbb{R}$, which takes $\lambda \in \Lambda \subseteq \mathbb{R}$ as an additional input, and is trained such that at inference time $h_\theta(\cdot, \lambda')$ recovers the classifier obtained by minimising $\mathcal{L}_{\lambda'}$.

Recall that we are interested in minimising the family of losses $\mathcal{L}_\lambda$, parameterized by $\lambda \in \Lambda$ (Eq. (2)). Instead of fixing $\lambda$, YOTO solves an optimisation problem where the parameter $\lambda$ is sampled from a distribution $P_\lambda$. As a result, during training the model observes many different values of $\lambda$ and learns to optimise the loss $\mathcal{L}_\lambda$ for all of them simultaneously. At inference time, the model can be conditioned on a chosen parameter value $\lambda'$ and recovers the model trained to optimise $\mathcal{L}_{\lambda'}$. The loss being minimised can thus be expressed as follows:

$$\arg\min_{h_\theta : \mathcal{X} \times \Lambda \to \mathbb{R}} \mathbb{E}_{\lambda \sim P_\lambda} \left[ \mathbb{E}[\mathcal{L}_{\text{CE}}(h_\theta(X, \lambda), Y)] + \lambda \, \mathcal{L}_{\text{fair}}(h_\theta(\cdot, \lambda)) \right]. \tag{4}$$

Having trained a YOTO model, the trade-off curve $\tau_{\text{fair}}^*(\psi)$ can be approximated by simply plugging in the values of $\lambda$ at inference time and thus avoiding additional training. From a theoretical point of view, Dosovitskiy & Djolonga (2020) prove that given a large enough model capacity, the above optimisation problem is equivalent to optimising the loss $\mathcal{L}_\lambda$ separately for different values of $\lambda$ as the minimum for both losses are the same (Dosovitskiy & Djolonga, 2020, Proposition 1). In other words, under the assumption of large enough model capacity, training the loss-conditional YOTO model performs as well as the separately trained models while only requiring a single model. To be clear, although the model capacity assumption might be hard to verify in practice, our experimental section has shown that the trade-off curves estimates $\widehat{\tau_{\text{fair}}^*(\psi)}$ obtained using YOTO models are consistent with the ones obtained using separately trained models.

It should be noted, as is common in optimization problems, that the estimated trade-off curve $\widehat{\tau_{\text{fair}}^*(\psi)}$ may not align precisely with the true trade-off curve $\tau_{\text{fair}}^*(\psi)$. This discrepancy originates from two key factors. Firstly, the limited size of the training and evaluation datasets influences the estimation of $\widehat{\tau_{\text{fair}}^*(\psi)}$. Secondly, we opt for a computationally tractable loss function instead of tackling the original constrained optimization problem, as stated in Eq. (1). To account for the estimation errors in $\widehat{\tau_{\text{fair}}^*(\psi)}$, we next show how YOTO model $h_\theta : \mathcal{X} \times \Lambda \to \mathbb{R}$ can be used to construct confidence intervals, designed to contain the true trade-off curve $\tau_{\text{fair}}^*(\psi)$ with high probability.

## 3.2 Step2: Constructing confidence intervals

In this section, we outline how to construct confidence intervals (CIs) for the optimal trade-off curve $\tau_{\text{fair}}^*(\psi)$ defined in Eq. (1). Specifically, we assume access to a held-out *calibration* dataset $\mathcal{D}_{\text{cal}} := \{(X_i, A_i, Y_i)\}_i$ which is disjoint from the training data. Given a level $\alpha \in [0, 1]$, we construct CIs $\Gamma_{\text{fair}}^\alpha \subseteq [0, 1]$ using $\mathcal{D}_{\text{cal}}$, which provide probabilistic guarantees of the form:

$$\mathbb{P}(\tau_{\text{fair}}^*(\Psi) \in \Gamma_{\text{fair}}^\alpha) \geq 1 - \alpha. \tag{5}$$

Here, it is important to note that $\Psi \in [0, 1]$ are random variables obtained from the calibration data $\mathcal{D}_{\text{cal}}$, and the guarantee in Eq. (5) holds marginally over $\Psi$ and $\Gamma_{\text{fair}}^\alpha$. We emphasize that our results in this section do not rely on a specific model class and for the sake of generality, we will outline our methodology in terms of general classifiers $h$ first and subsequently establish how the results apply to YOTO models specifically. Before we construct intervals on $\tau_{\text{fair}}^*$, our methodology involves first constructing CIs using $\mathcal{D}_{\text{cal}}$ on accuracy $\text{acc}(h)$ and fairness violation $\Phi_{\text{fair}}(h)$ for a given model $h$, denoted as $C_{\text{acc}}^\alpha(h)$ and $C_{\text{fair}}^\alpha(h)$ respectively, which satisfy:

$$\mathbb{P}(\text{acc}(h) \in C_{\text{acc}}^\alpha(h)) \geq 1 - \alpha \quad \text{and} \quad \mathbb{P}(\Phi_{\text{fair}}(h) \in C_{\text{fair}}^\alpha(h)) \geq 1 - \alpha.$$

One possible way to construct these confidence intervals involves using assumption-light concentration inequalities such as Hoeffding's inequality. To be more concrete for $\text{acc}(h)$:

**Lemma 3.1** (Hoeffding's inequality). *Given a classifier $h : \mathcal{X} \to \mathcal{Y}$, we have that,*

$$\mathbb{P}\left(\text{acc}(h) \in \left[\widehat{\text{acc}(h)} - \delta, \widehat{\text{acc}(h)} + \delta\right]\right) \geq 1 - \alpha,$$

*where* $\widehat{\text{acc}(h)} := \sum_{(X_i, A_i, Y_i) \in \mathcal{D}_{\text{cal}}} \frac{\mathbb{1}(h(X_i) = Y_i)}{|\mathcal{D}_{\text{cal}}|}$ *and* $\delta := \sqrt{\frac{1}{2|\mathcal{D}_{\text{cal}}|} \log\left(\frac{2}{\alpha}\right)}$.

Lemma 3.1 illustrates that we can use Hoeffding's inequality to construct confidence interval $C_{\text{acc}}^\alpha(h) = [\widehat{\text{acc}(h)} - \delta, \widehat{\text{acc}(h)} + \delta]$ on $\text{acc}(h)$ such that the true $\text{acc}(h)$ will lie inside the interval with probability $1 - \alpha$. Analogously, we are also able to establish confidence intervals for fairness violations, denoted as $\Phi_{\text{fair}}(h)$, albeit subject to certain nuanced challenges. Due to space constraints, we have detailed the specific methodology for the fairness violation in Appendix B. Next, we outline how to derive confidence intervals for the minimum achievable fairness $\tau_{\text{fair}}^*$, satisfying Eq. (5).

### 3.2.1 Upper confidence intervals

Here, we outline how to obtain one-sided upper confidence intervals on the minimum attainable accuracy constrained fairness $\tau_{\text{fair}}^*(\Psi)$ of the form $\Gamma_{\text{fair}}^\alpha = [0, U_{\text{fair}}^\alpha]$, which satisfies the probabilistic guarantee in Eq. (5). To this end, given a classifier $h \in \mathcal{H}$, our methodology involves constructing one-sided lower CI on the accuracy $\text{acc}(h)$ and upper CI on the fairness violation $\Phi_{\text{fair}}(h)$. We make this concrete in the following result:

**Proposition 3.2.** *Given classifier $h \in \mathcal{H}$, let $L_{\text{acc}}^\alpha, U_{\text{fair}}^\alpha \in [0, 1]$ be such that*

$$\mathbb{P}(\text{acc}(h) \geq L_{\text{acc}}^\alpha) \geq 1 - \alpha/2 \quad \text{and} \quad \mathbb{P}(\Phi_{\text{fair}}(h) \leq U_{\text{fair}}^\alpha) \geq 1 - \alpha/2.$$

*Then,* $\mathbb{P}\left(\tau_{\text{fair}}^*(L_{\text{acc}}^\alpha) \leq U_{\text{fair}}^\alpha\right) \geq 1 - \alpha$.

Recall our original goal of constructing a one-sided confidence interval on the minimum attainable fairness violation for models with accuracy at least $\psi$ (i.e., $\tau_{\text{fair}}^*(\psi)$). Proposition 3.2 can be used to construct such intervals by first finding a model $h \in \mathcal{H}$ for which the lower CI on accuracy, $L_{\text{acc}}^\alpha$, satisfies $L_{\text{acc}}^\alpha \geq \psi$. Then, since $\tau_{\text{fair}}^*$ is a monotonically increasing function, we have that $\tau_{\text{fair}}^*(\psi) \leq \tau_{\text{fair}}^*(L_{\text{acc}}^\alpha)$ and since $U_{\text{fair}}^\alpha$ is an upper CI for $\tau_{\text{fair}}^*(L_{\text{acc}}^\alpha)$, it follows that $U_{\text{fair}}^\alpha$ can also serve as an upper CI for $\tau_{\text{fair}}^*(\psi)$. Proposition 3.2 can straightforwardly be applied to YOTO models:

**Corollary 3.3.** *Let $h_\theta : \mathcal{X} \times \Lambda \to \mathbb{R}$ be a YOTO model, and for $\lambda_0 \in \Lambda$, let $L_{\mathrm{acc}}^\alpha, U_{\mathrm{fair}}^\alpha$ be s.t.*

$$\mathbb{P}(\mathrm{acc}(h_\theta(\cdot, \lambda_0)) \geq L_{\mathrm{acc}}^\alpha) \geq 1 - \alpha/2 \quad \text{and} \quad \mathbb{P}(\Phi_{\mathrm{fair}}(h_\theta(\cdot, \lambda_0)) \leq U_{\mathrm{fair}}^\alpha) \geq 1 - \alpha/2.$$

*Then,* $\mathbb{P}(\tau_{\mathrm{fair}}^*(L_{\mathrm{acc}}^\alpha) \leq U_{\mathrm{fair}}^\alpha) \geq 1 - \alpha.$

Finally, it is important to note that the Proposition 3.2 and Corollary 3.3 do not rely on any assumptions regarding the optimality of the trained classifiers. This means that the upper confidence intervals will remain valid even if the classifier $h$ is not trained well (and hence achieves sub-optimal accuracy-fairness trade-offs), although in such cases the confidence interval may be conservative. Next, we show how to construct one-sided lower confidence intervals on $\tau_{\mathrm{fair}}^*(\psi)$.

### 3.2.2 Lower confidence intervals

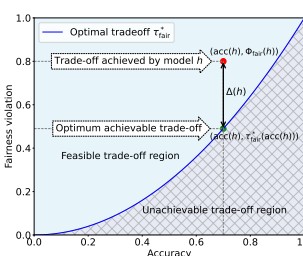

Figure 2: Visual representation of the difference between the ground truth optimal and model achieved trade-offs.

Here, we explain at an intuitive level why obtaining lower confidence intervals on $\tau_{\mathrm{fair}}^*(\psi)$ is more challenging than obtaining upper confidence intervals. Suppose that $h \in \mathcal{H}$ is such that $\mathrm{acc}(h) = \psi$, then since $\tau_{\mathrm{fair}}^*$ denotes the minimum attainable fairness violation (Eq. (1)), we have that $\tau_{\mathrm{fair}}^*(\psi) \leq \Phi_{\mathrm{fair}}(h)$. Therefore, any valid upper confidence interval on $\Phi_{\mathrm{fair}}(h)$ will also be valid for $\tau_{\mathrm{fair}}^*(\psi)$. However, a lower bound on $\Phi_{\mathrm{fair}}(h)$ cannot be used as a lower bound for the minimum achievable fairness $\tau_{\mathrm{fair}}^*(\psi)$ in general. Constructing a lower CI will require assumptions on how close the fairness violation $\Phi_{\mathrm{fair}}(h)$ is to the minimum achievable fairness violation $\tau_{\mathrm{fair}}^*(\psi)$ (i.e., $\Delta(h)$ term in Figure 2). We make this concrete below by constructing lower CIs on $\tau_{\mathrm{fair}}^*(\psi)$ which are unknown in general unless we make assumptions on the optimality of fairness violation $\Phi_{\mathrm{fair}}(h)$ (i.e., if we assume $\Delta(h) \leq c$ for some $c$).

**Proposition 3.4.** *Suppose that, for a given classifier $h \in \mathcal{H}$,*

$$\mathbb{P}(\mathrm{acc}(h) \leq U_{\mathrm{acc}}^\alpha) \geq 1 - \alpha/2 \quad \text{and} \quad \mathbb{P}(\Phi_{\mathrm{fair}}(h) \geq L_{\mathrm{fair}}^\alpha) \geq 1 - \alpha/2.$$

*Then,* $\mathbb{P}\left(\tau_{\mathrm{fair}}^*(U_{\mathrm{acc}}^\alpha) \geq L_{\mathrm{fair}}^\alpha - \Delta(h)\right) \geq 1 - \alpha$, *where* $\Delta(h) \coloneqq \Phi_{\mathrm{fair}}(h) - \tau_{\mathrm{fair}}^*(\mathrm{acc}(h)) \geq 0.$

Proposition 3.4 provides the guarantee in Eq. (5) with $\Psi = U_{\mathrm{acc}}^\alpha$. Like Proposition 3.2, this result shows that if the goal is to construct lower confidence intervals on $\tau_{\mathrm{fair}}^*(\psi)$ and we obtain that $\psi \geq U_{\mathrm{acc}}^\alpha$, then using the monotonicity of $\tau_{\mathrm{fair}}^*$ we have that $\tau_{\mathrm{fair}}^*(\psi) \geq \tau_{\mathrm{fair}}^*(U_{\mathrm{acc}}^\alpha)$. Therefore since $L_{\mathrm{fair}}^\alpha$ is a lower confidence interval for $\tau_{\mathrm{fair}}^*(U_{\mathrm{acc}}^\alpha)$, it also serves as a lower CI for $\tau_{\mathrm{fair}}^*(\psi)$. Analogously to Corollary 3.3, this result can be directly applied to YOTO (see Corollary C.1).

Recall that $\Delta(h)$ quantifies how 'far' the fairness loss of classifier $h$ is from the minimum attainable fairness loss $\tau_{\mathrm{fair}}^*(\mathrm{acc}(h))$ and is an unknown quantity in general (see Figure 2). One practical choice which allows us to obtain the lower confidence intervals exactly is to assume that $\Delta(h) = 0$, i.e. that the model $h$ achieves the lowest attainable fairness loss $\tau_{\mathrm{fair}}^*(\mathrm{acc}(h))$. In this case, the confidence intervals quantify the uncertainty in the trade-off $\tau_{\mathrm{fair}}^*$ arising due to finite calibration data.

However, the assumption $\Delta(h) = 0$ may be considered too restrictive as the model $h$ will in practice not achieve the optimal accuracy-fairness trade-off $\tau_{\mathrm{fair}}^*$. To remediate this, we employ sensitivity analysis techniques to incorporate any belief on plausible values for $\Delta(h)$. This allows us to construct CIs which not only incorporate finite sample uncertainty from calibration data, but also account for the possible sub-optimality in the fairness trade-offs achieved by $h$. We provide more details in Appendix C. In addition to this, in Appendix C.2, we show that under certain mild assumptions on the model $h$, we can obtain probabilistic bounds on $\Delta(h)$ which show that as the number of training data increases, the $\Delta(h)$ term will tend to 0 with high probability.

### 3.3 Handling Scarce Sensitive Attributes

Next, we consider the case where sensitive attributes $A$ are accessible for only a small subset of the calibration dataset $\mathcal{D}_{\mathrm{cal}}$, and constructing reliable confidence intervals (CIs) for $\Phi_{\mathrm{fair}}(h)$ becomes challenging. Intuitively, taking a closer look at Lemma 3.1, we can see that the bigger the $|\mathcal{D}_{\mathrm{cal}}|$ dataset, the tighter the bounds are. Hence, when we only have a few data points for which we have

the sensitive attributes, constructing CIs only using a small subset of $\mathcal{D}_{\text{cal}}$ with available sensitive attributes can lead to highly conservative intervals. One way to fix this issues would be to predict the missing $A$ values using a surrogate model $f_{\mathcal{A}} : \mathcal{X} \to \mathcal{A}$. However, in this case, the estimated value of fairness violations can be significantly biased, particularly if $f_{\mathcal{A}}$ has low accuracy.

Hence, inspired by prediction-powered inference (Angelopoulos et al., 2023), we introduce a method that effectively combines data from both subsets of $\mathcal{D}_{\text{cal}}$, i.e. data with actual and predicted sensitive attributes to derive tighter and more accurate CIs, even when the majority of $A$ values are absent. Our methodology focuses on the discrepancy between fairness violations assessed with actual and surrogate sensitive attributes. On a high level, the key idea is to adjust for the potential bias introduced by surrogate predictions $f_{\mathcal{A}}(X)$, using the small amount of data with true sensitive attributes. We empirically confirm in Section 5 that our proposed intervals are (i) tighter than those obtained using only the small subset with true $A$ values, and (ii) more well-calibrated than the CIs obtained by imputing missing sensitive attributes $A$ with predicted sensitive attributes $f_{\mathcal{A}}(X)$. Due to space constraints, the exact details of our methodology are provided in Appendix D.

# 4 RELATED WORKS

In-processing methods for mitigating fairness violations commonly introduce constraints or regularization terms to the optimization objective. For instance, Agarwal et al. (2018) maximizes model accuracy while constraining fairness violations. However, given the data-dependent nature of accuracy-fairness trade-offs, setting a universal fairness threshold may not be suitable. Various other regularization approaches (Wei & Niethammer, 2022; Olfat & Mintz, 2020; Bendekgey & Sudderth, 2021; Donini et al., 2018; Zafar et al., 2015; 2017; 2019) also exist, but they often necessitate training multiple models, making them computationally intensive.

Alternative fairness strategies include learning 'fair' data representations (Zemel et al., 2013; Louizos et al., 2017; Lum & Johndrow, 2016), or pre-processing data through re-weighting based on sensitive attributes (Grover et al., 2019; Kamiran & Calders, 2011). These, however, provide limited control over accuracy-fairness trade-offs. Post-processing methods (Hardt et al., 2016; Wei et al., 2020) enforce fairness after training but can lead to other forms of unfairness (EEOC, 1979). Beyond fairness, Lin et al. (2020) applies YOTO to multi-task learning. Our work is unique, being the first to adapt YOTO to fairness and the first in fairness to construct valid CIs on the optimal trade-off curve, considering finite-sample estimation.

# 5 EXPERIMENTS

Having established the theoretical guarantees and bounds for the confidence intervals surrounding the fairness trade-off curve, denoted as $\tau_{\text{fair}}^*(\psi)$, we now proceed to empirically validate these intervals across diverse datasets. These datasets span from tabular (e.g., `Adult` and `COMPAS` ), to image-based (e.g., `CelebA`), and natural language processing datasets (e.g., `Jigsaw`). Recall that, our approach involves a two-step methodology: initial estimation of the trade-off curve via the YOTO model, followed by the construction of confidence intervals through a separate calibration dataset, $\mathcal{D}_{\text{cal}}$, which will contain the ground truth trade-off curve with high probability.

To evaluate our methodology, we implement a suite of baseline algorithms in the fair machine learning literature. This includes state-of-the-art in-processing techniques such as regularization-based approaches (Bendekgey & Sudderth, 2021), as well as the popular reduction methods (Agarwal et al., 2018). Additionally, we also conduct experiments using adversarial techniques aimed at fair representation learning (Zhang et al., 2018). Finally, to further substantiate the universal applicability of our proposed confidence intervals, we show that they are effective across the three most prominent fairness metrics: Demographic Parity (DP), Equalized Odds (EO), and Equalized Opportunity (EOP). We provide additional experimental details and results in Appendix E.

## 5.1 RESULTS

Figure 3 shows the results for different datasets and fairness violations, obtained using a calibration dataset $\mathcal{D}_{\text{cal}}$ of size 2000. For each dataset, we construct 4 confidence intervals that serve as the upper and lower bounds on the optimal accuracy-fairness trade-off curve. These intervals are computed

at a 95% confidence level using various methodologies, including 1) Hoeffding's, 2) Bernstein's inequalities which both offer finite sample guarantees as well as, 3) bootstrapping (Efron, 1979), and 4) asymptotic intervals based on the Central Limit Theorem (Le Cam, 1986) which are valid asymptotically in the number of calibration data $\mathcal{D}_{\text{cal}}$. The findings of our experiments are as follows:

**Trade-off curves are data dependent:** Firstly, the results in Figure 3 confirm that the accuracy-fairness trade-offs can vary significantly across the datasets. For example, achieving near-perfect fairness (i.e. $\Phi_{\text{fair}}(h) \approx 0$) seems significantly easier for the Jigsaw dataset than the COMPAS dataset, even as the accuracy increases. Likewise, we observe that for Adult and COMPAS datasets the optimal DP increases gradually with increasing accuracy (smoothly), whereas for the CelebA dataset, the increase is sharp once the accuracy increases above 90% (i.e., for CelebA the additional accuracy beyond 90% level comes at a significant cost in terms of fairness violations). These trade-off discrepancies across the datasets support our argument that using a universal fairness threshold across datasets may be too restrictive as in Agarwal et al. (2018), and our methodology provides more dataset-specific insights about the entire trade-off curve instead.

**CIs contain the empirical trade-offs:** The CIs presented in Figure 3, use $\Delta(h) = 0$ when constructing the lower CIs. Despite this choice of $\Delta(h)$, all four of our proposed confidence intervals successfully encapsulate the empirical accuracy-fairness trade-offs for the majority of the SOTA baselines (separate, logsig, linear, reductions, adversary) examined. While the CIs obtained using Hoeffding's inequality, are comparatively conservative, the asymptotic, bootstrap and Bernstein CIs are relatively tight and informative in most cases. Note that our intervals are designed to align with the optimal trade-off curve and therefore, any methodology whose trade-off lies above our upper bound is likely suboptimal, suggesting that alternative approaches may offer improved trade-offs.

**YOTO trade-offs are consistent with SOTA:** We observe that the empirical trade-offs obtained using the YOTO models align well with most of the SOTA baselines considered, while avoiding the computational cost of training multiple models. This shows that the YOTO model successfully manages to approximate the optimal accuracy-fairness trade-off curves, in most cases as well as (and in some cases, better than) the baselines considered. For example, for COMPAS dataset, the YOTO trade-off curve is consistent with the baseline results, whereas for the Jigsaw dataset, the YOTO model achieves better fairness-accuracy trade-offs than most baselines (especially for EOP results).

**YOTO leads to a smoother trade-off curve than baselines:** We observe that baselines utilizing the reductions, regularization and adversarial approaches — which involve training multiple, independently trained models — not only impose computational burden but also yield empirical trade-offs with high variance as accuracy increases (see Jigsaw results in Figure 3, for example). This behaviour starkly contrasts with the smooth variations exhibited by our YOTO-generated trade-off curves along the accuracy axis. This leads to CIs which vary smoothly with accuracy and allows us to reliably obtain an acceptable range of fairness violations for specific accuracy thresholds.

### 5.1.1 SCARCE SENSITIVE ATTRIBUTES

As detailed in Section 3.3, we also analyze scenarios where access to the ground truth sensitive attributes $A$ is scarce within $\mathcal{D}_{\text{cal}}$. Figure 4 displays the CIs using three methodologies, evaluating both YOTO and separately trained models using all available $A$. In Figure 4a, using only data points with true sensitive attributes yields conservative intervals due to reduced calibration data usage. Conversely, Figure 4b, which imputes missing attributes with predicted values $f_{\mathcal{A}}(X)$, produces tighter, yet miscalibrated intervals, due to a 75% accuracy of $f_{\mathcal{A}}(X)$ causing bias in fairness estimation. Figure 4c employs our combined datasets method, leading to tighter intervals compared to Figure 4a and more well-calibrated compared to Figure 4b. This illustrates our approach's ability to account for the prediction error in $f_{\mathcal{A}}$. Comprehensive ablations are provided in Appendix D.

## 6 DISCUSSION AND LIMITATIONS

In this work, we propose a novel and computationally efficient approach to capture the fairness-accuracy trade-offs inherent to individual datasets, backed by sound statistical guarantees. Our proposed methodology enables a nuanced and dataset-specific understanding of the fairness-accuracy trade-offs. It does so by obtaining confidence intervals on the accuracy-fairness trade-off, leveraging the computational benefits of the You-Only-Train-Once (YOTO) framework (Dosovitskiy &

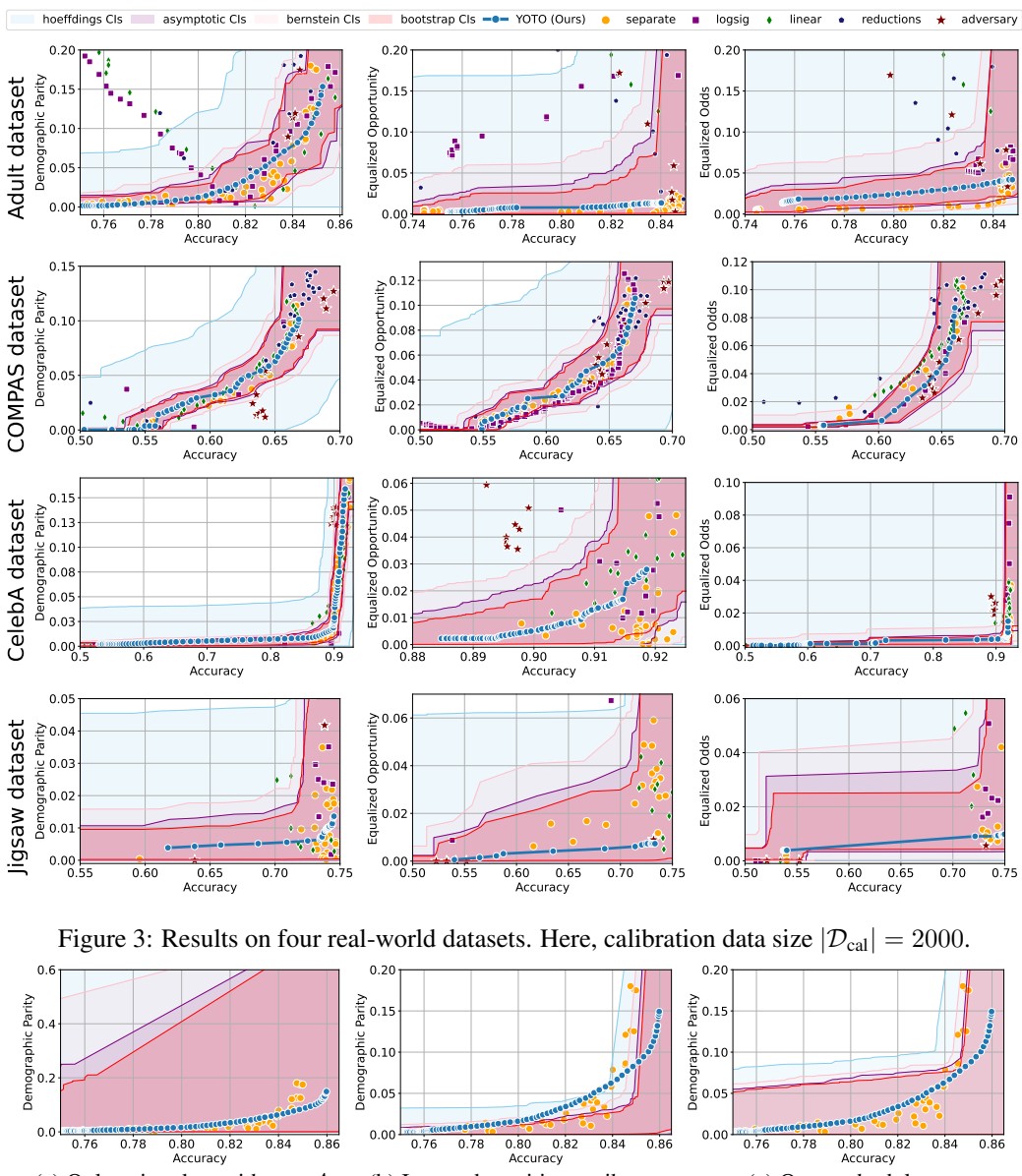

Figure 3: Results on four real-world datasets. Here, calibration data size $|\mathcal{D}_{\text{cal}}| = 2000$.

(a) Only using data with true $A$    (b) Imputed sensitive attributes    (c) Our methodology

Figure 4: CIs constructed for Adult dataset in the setting where sensitive attributes $A$ are only available for 50 out of the 2,550 calibration data points and $\text{acc}(f_{\mathcal{A}}) = 75\%$.

Djolonga, 2020). This empowers practitioners with the capability to, at inference time, specify desired accuracy levels and promptly receive corresponding admissible fairness violation ranges. By eliminating the need for repetitive model training, we significantly streamline the process of understanding and interpreting fairness-accuracy trade-offs tailored to individual datasets.

**Limitations** Despite the evident merits of our approach, it also has some potential limitations. Firstly, our methodology requires distinct datasets for both training and calibration, posing difficulties in situations with limited data resources. Under such constraints, the YOTO model might not capture the optimal fairness-accuracy trade-off, and moreover, the resulting confidence intervals could be overly conservative. Secondly, our lower confidence intervals incorporate an unknown term $\Delta(h)$. While we propose sensitivity analysis strategies for approximating this term and delve deeper into its potential bounds under certain mild assumptions in Appendix C.2, a more exhaustive understanding remains an open research question. Exploring ways to derive rigorous and informative upper bounds for the $\Delta(h)$ under weaker conditions is a promising avenue for future investigations.

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

## A  PROOFS

*Proof of Lemma 3.1.*  This lemma is a straightforward application of Heoffding's inequality.  □

*Proof of Proposition 3.2.*  Using a straightforward application union bounds, we get that
$$\mathbb{P}(\mathrm{acc}(h) \geq L^\alpha_{\mathrm{acc}}, \Phi_{\mathrm{fair}}(h) \leq U^\alpha_{\mathrm{fair}}) \geq 1 - \mathbb{P}(\mathrm{acc}(h) < L^\alpha_{\mathrm{acc}}) - \mathbb{P}(\Phi_{\mathrm{fair}}(h) > U^\alpha_{\mathrm{fair}})$$
$$\geq 1 - \alpha/2 - \alpha/2 = 1 - \alpha.$$
Using the definition of the optimal fairness-accuracy trade-off $\tau^*_{\mathrm{fair}}$, we get that the event

$$\{\mathrm{acc}(h) \geq L^\alpha_{\mathrm{acc}}, \Phi_{\mathrm{fair}}(h) \leq U^\alpha_{\mathrm{fair}}\} \quad \text{implies,} \quad \left\{ \underbrace{\min\{\Phi_{\mathrm{fair}}(h') \mid h' \in \mathcal{H}, \mathrm{acc}(h') \geq L^\alpha_{\mathrm{acc}}\}}_{\tau^*_{\mathrm{fair}}(L^\alpha_{\mathrm{acc}})} \leq U^\alpha_{\mathrm{fair}} \right\}.$$

From this, it follows that
$$\mathbb{P}(\tau^*_{\mathrm{fair}}(L^\alpha_{\mathrm{acc}}) \leq U^\alpha_{\mathrm{fair}}) \geq \mathbb{P}(\mathrm{acc}(h) \geq L^\alpha_{\mathrm{acc}}, \Phi_{\mathrm{fair}}(h) \leq U^\alpha_{\mathrm{fair}}) \geq 1 - \alpha.$$
□

*Proof of Corollary 3.3.*  This corollary follows straightforwardly from Proposition 3.2.  □

*Proof of Proposition 3.4.*  Using an application of union bounds, we get that
$$\mathbb{P}(\mathrm{acc}(h) \leq U^\alpha_{\mathrm{acc}}, \Phi_{\mathrm{fair}}(h) \geq L^\alpha_{\mathrm{fair}}) \geq 1 - \mathbb{P}(\mathrm{acc}(h) > U^\alpha_{\mathrm{acc}}) - \mathbb{P}(\Phi_{\mathrm{fair}}(h) < L^\alpha_{\mathrm{fair}})$$
$$\geq 1 - \alpha/2 - \alpha/2 = 1 - \alpha.$$
Then, using the fact that $\Delta(h) = \Phi_{\mathrm{fair}}(h) - \tau^*_{\mathrm{fair}}(\mathrm{acc}(h))$, we get that
$$1 - \alpha \leq \mathbb{P}(\mathrm{acc}(h) \leq U^\alpha_{\mathrm{acc}}, \Phi_{\mathrm{fair}}(h) \geq L^\alpha_{\mathrm{fair}}) = \mathbb{P}(\mathrm{acc}(h) \leq U^\alpha_{\mathrm{acc}}, \tau^*_{\mathrm{fair}}(\mathrm{acc}(h)) + \Delta(h) \geq L^\alpha_{\mathrm{fair}})$$
$$\leq \mathbb{P}(\mathrm{acc}(h) \leq U^\alpha_{\mathrm{acc}}, \tau^*_{\mathrm{fair}}(U^\alpha_{\mathrm{acc}}) + \Delta(h) \geq L^\alpha_{\mathrm{fair}})$$
$$\leq \mathbb{P}(\tau^*_{\mathrm{fair}}(U^\alpha_{\mathrm{acc}}) \geq L^\alpha_{\mathrm{fair}} - \Delta(h)),$$
where in the second last inequality above, we use the fact that $\tau^*_{\mathrm{fair}} : [0, 1] \to [0, 1]$ is a monotonically increasing function.  □

## B  CONSTRUCTING THE CONFIDENCE INTERVALS ON $\Phi_{\mathrm{fair}}(h)$

In this section, we outline methodologies of obtaining confidence intervals for a fairness violation $\Phi_{\mathrm{fair}}$. Specifically, given a model $h \in \mathcal{H}$, with $h : \mathcal{X} \to \mathcal{Y}$ and $\alpha \in (0, 1)$, we outline how to find $C^\alpha_{\mathrm{fair}}$ which satisfies,
$$\mathbb{P}(\Phi_{\mathrm{fair}}(h) \in C^\alpha_{\mathrm{fair}}) \geq 1 - \alpha. \tag{6}$$
Similar to Agarwal et al. (2018) we express the fairness violation $\Phi_{\mathrm{fair}}$ as:

$$\Phi_{\mathrm{fair}}(h) = |\Phi^\pm_{\mathrm{fair}}(h)| \quad \text{where,} \quad \Phi^\pm_{\mathrm{fair}}(h) := \sum_{j=1}^m \underbrace{\mathbb{E}[g_j(X, A, Y, h(X)) \mid \mathcal{E}_j]}_{=: \Phi_j}$$

where $m \geq 1$, $g_j$ are some known functions and $\mathcal{E}_j$ are events with positive probability defined with respect to $(X, A, Y)$. For example, when considering the demographic parity (DP), i.e. $\Phi_{\mathrm{fair}} = \Phi_{\mathrm{DP}}$, we have $m = 2$, with $g_1(X, A, Y, h(X)) = h(X)$, $\mathcal{E}_1 = \{A = 1\}$, $g_2(X, A, Y, h(X)) = -h(X)$ and $\mathcal{E}_2 = \{A = 0\}$. Moreover, as shown in Agarwal et al. (2018), the commonly used fairness metrics like Equalized Odds (EO) and Equalized Opportunity (EOP) can also be expressed in similar forms.

Our methodology of constructing CIs on $\Phi_{\mathrm{fair}}(h)$ involves first constructing intervals $C^{\alpha,\pm}_{\mathrm{fair}}$ on $\Phi^\pm_{\mathrm{fair}}(h)$ satisfying:
$$\mathbb{P}(\Phi^\pm_{\mathrm{fair}}(h) \in C^{\alpha,\pm}_{\mathrm{fair}}) \geq 1 - \alpha. \tag{7}$$
Once we have a $C^{\alpha,\pm}_{\mathrm{fair}}$, the confidence interval $C^\alpha_{\mathrm{fair}}$ satisfying Eq. (6) can simply be constructed as:
$$C^\alpha_{\mathrm{fair}} = \{|x| \; : \; x \in C^{\alpha,\pm}_{\mathrm{fair}}\}.$$

In what follows, we outline two different ways of constructing the confidence intervals $C^{\alpha,\pm}_{\mathrm{fair}}$ on $\Phi^\pm_{\mathrm{fair}}(h)$ satisfying Eq. (7).

## B.1 Separately constructing CIs on $\Phi_j$

One way to obtain intervals on $\Phi_{\text{fair}}^{\pm}(h)$ would be to separately construct confidence intervals on $\Phi_j$, denoted by $C_j^{\alpha}$, which satisfies the joint guarantee

$$\mathbb{P}\left(\cap_{j=1}^m \{\Phi_j \in C_j^{\alpha}\}\right) \geq 1 - \alpha. \tag{8}$$

Given such set of confidence intervals $\{C_j^{\alpha}\}_{j=1}^m$ which satisfy Eqn. Eq. (8), we can obtain the confidence intervals on $\Phi_{\text{fair}}^{\pm}(h)$ by using the fact that

$$\mathbb{P}\left(\Phi_{\text{fair}}^{\pm}(h) \in \sum_{i=1}^m C_j^{\alpha}\right) \geq 1 - \alpha.$$

Where, the notation $\sum_{i=1}^m C_j^{\alpha}$ denotes the set $\{\sum_{i=1}^m x_i \; : \; x_i \in C_i^{\alpha}\}$. One naïve way to obtain such $\{C_j^{\alpha}\}_{j=1}^m$ which satisfy Eq. (8) is to use the union bounds, i.e., if $C_j^{\alpha}$ are chosen such that

$$\mathbb{P}(\Phi_j \in C_j^{\alpha}) \leq 1 - \alpha/m,$$

then, we have that

$$\mathbb{P}\left(\cap_{j=1}^m \{\Phi_j \in C_j^{\alpha}\}\right) = 1 - \mathbb{P}(\cup_{j=1}^m \{\Phi_j \in C_j^{\alpha}\}^c)$$

$$\leq 1 - \sum_{i=1}^m \mathbb{P}(\{\Phi_j \in C_j^{\alpha}\}^c)$$

$$\leq 1 - \sum_{i=1}^m (1 - (1 - \alpha/m)) = 1 - \alpha.$$

Here, for an event $\mathcal{E}$, we use $\mathcal{E}^c$ to denote the complement of the event. This methodology therefore reduces the problem of finding confidence intervals on $\Phi_{\text{fair}}^{\pm}(h)$ to finding confidence intervals on $\Phi_j$ for $j \in \{1, \ldots, m\}$. Now note that $\Phi_j$ are all expectations and we can use standard methodologies to construct confidence intervals on an expectation. We explicitly outline how to do this in Section B.3.

**Remark** The methodology outlined above provides confidence intervals with valid finite sample coverage guarantees. However, this may come at the cost of more conservative confidence intervals. One way to obtain less conservative confidence intervals while retaining the coverage guarantees would be to consider alternative ways of obtaining confidence intervals which do not require constructing the CIs separately on $\Phi_j$. We outline one such methodology in the next section.

## B.2 Using subsampling to construct the CIs on $\Phi_{\text{fair}}^{\pm}$ directly

Here, we outline how we can avoid having to use union bounds when constructing the confidence intervals on $\Phi_{\text{fair}}^{\pm}$. Let $\mathcal{D}_j$ denote the subset of data $\mathcal{D}_{\text{cal}}$, for which the event $\mathcal{E}_j$ is true. In the case where the events $\mathcal{E}_j$ are all mutually exclusive and hence $\mathcal{D}_j$ are all disjoint subsets of data (which is true for DP, EO and EOP), we can also construct these intervals by randomly sampling without replacement datapoints $(x_i^{(j)}, a_i^{(j)}, y_i^{(j)})$ from $\mathcal{D}_j$ for $i \leq l := \min_{k \leq m} |\mathcal{D}_k|$. We use the fact that

$$\widehat{\Phi_{\text{fair}}^{\pm}}(h) = \frac{1}{l} \sum_{i=1}^l \sum_{j=1}^m g_j(x_i^{(j)}, a_i^{(j)}, y_i^{(j)}, h(x_i^{(j)}))$$

is an unbiased estimator of $\Phi_{\text{fair}}^{\pm}(h)$. Moreover, since $\mathcal{D}_j$ are all disjoint datasets, the datapoints $(x_i^{(j)}, a_i^{(j)}, y_i^{(j)})$ are all independent across different values of $j$, and therefore, $\sum_{j=1}^m g_j(x_i^{(j)}, a_i^{(j)}, y_i^{(j)}, h(x_i^{(j)}))$ are i.i.d.. In other words,

$$\widehat{\Phi_{\text{fair}}^{\pm}}(h) = \frac{1}{l} \sum_{i=1}^l \phi_i \quad \text{where,} \quad \phi_i := \sum_{j=1}^m g_j(x_i^{(j)}, a_i^{(j)}, y_i^{(j)}, h(x_i^{(j)}))$$

and $\phi_i$ are all i.i.d. samples and unbiased estimators of $\Phi_{\text{fair}}^{\pm}(h)$. Therefore, like in the previous section, our problem reduces to constructing CIs on an expectation term (i.e. $\Phi_{\text{fair}}^{\pm}(h)$), using i.i.d. unbiased samples (i.e. $\phi_i$) and we can use standard methodologies to construct these intervals.

**Benefit of this methodology**    This methodology no longer requires us to separately construct confidence intervals over $\Phi_j$ and combine them using union bounds (for example). Therefore, intervals obtained using this methodology may be less conservative than those obtained by separately constructing confidence intervals over $\Phi_j$.

**Limitation of this methodology**    For each subset of data $\mathcal{D}_j$, we can use at most $l := \min_{k \leq m} |\mathcal{D}_k|$ data points to construct the confidence intervals. Therefore, in cases where $l$ is very small, we may end up discarding a big proportion of the calibration data which could in turn lead to loose intervals.

### B.3    Constructing CIs on expectations

Here, we outline some standard techniques used to construct CIs on the expectation of a random variable. These techniques can then be used to construct CIs on $\Phi_{\text{fair}}(h)$ (using either of the two methodologies outlined above) as well as on $\text{acc}(h)$. In this section, we restrict ourselves to constructing lower CIs. Upper CIs can be constructed analogously.

Given dataset $\{Z_i : 1 \leq i \leq n\}$, our goal in this section is to construct upper CIs on $\mathbb{E}[Z]$ which satisfies

$$\mathbb{P}(\mathbb{E}[Z] \leq U^\alpha) \geq 1 - \alpha.$$

**Hoeffding's inequality**    We can use Hoeffding's inequality to construct these intervals $U^\alpha$ as formalised in the following result:

**Lemma B.1** (Hoeffding's inequality). *Let $Z_i \in [0, 1]$, $1 \leq i \leq n$ be i.i.d. samples with mean $\mathbb{E}[Z]$. Then,*

$$\mathbb{P}\left( \mathbb{E}[Z] \leq \frac{1}{n} \sum_{i=1}^{n} Z_i + \sqrt{\frac{1}{2n} \log \frac{1}{\alpha}} \right) \geq 1 - \alpha.$$

**Bernstein's inequality**    Bernstein's inequality provides a powerful tool for bounding the tail probabilities of the sum of independent, bounded random variables. Specifically, for a sum $\sum_{i=1}^{n} Z_i$ comprised of $n$ independent random variables $Z_i$ with $Z_i \in [0, B]$, each with a maximum variance of $\sigma^2$, and for any $t > 0$, the inequality states that

$$\mathbb{P}\left( \mathbb{E}\left[ \sum_{i=1}^{n} Z_i \right] - \sum_{i=1}^{n} Z_i > t \right) \leq \exp\left( -\frac{t^2}{2\sigma^2 + \frac{2}{3}tB} \right),$$

where $B$ denotes an upper bound on the absolute value of each random variable. Re-arranging the above, we get that

$$\mathbb{P}\left( \mathbb{E}[Z] < \frac{1}{n} \left( \sum_{i=1}^{n} Z_i + t \right) \right) \geq 1 - \exp\left( -\frac{t^2}{2\sigma^2 + \frac{2}{3}tB} \right).$$

This allows us to construct upper CIs on $\mathbb{E}[Z]$.

**Central Limit Theorem**    The Central Limit Theorem (CLT) (Le Cam, 1986) serves as a cornerstone in statistics for constructing confidence intervals around sample means, particularly when the sample size is substantial. The theorem posits that, for a sufficiently large sample size, the distribution of the sample mean will closely resemble a normal (Gaussian) distribution, irrespective of the original population's distribution. This Gaussian nature of the sample mean empowers us to form confidence intervals for the population mean using the normal distribution's characteristics.

Given $Z_1, Z_2, \dots, Z_n$ as $n$ independent and identically distributed (i.i.d.) random variables with mean $\mu$ and variance $\sigma^2$, the sample mean $\bar{Z}$ approximates a normal distribution with mean $\mu$ and variance $\sigma^2/n$ for large $n$. An upper $(1 - \alpha)$ confidence interval for $\mu$ is thus:

$$U^\alpha = \bar{Z} + z_{\alpha/2} \frac{\sigma}{\sqrt{n}}$$

where $z_{\alpha/2}$ represents the critical value from the standard normal distribution corresponding to a cumulative probability of $1 - \alpha$.

**Bootstrap Confidence Intervals** Bootstrapping, introduced by Efron (1979), offers a non-parametric approach to estimate the sampling distribution of a statistic. The method involves repeatedly drawing samples (with replacement) from the observed data and recalculating the statistic for each resample. The resulting empirical distribution of the statistic across bootstrap samples forms the basis for confidence interval construction.

Given a dataset $Z_1, Z_2, \ldots, Z_n$, one can produce $B$ bootstrap samples by selecting $n$ observations with replacement from the original data. For each of these samples, the statistic of interest (for instance, the mean) is determined, yielding $B$ bootstrap estimates. An upper $(1 - \alpha)$ bootstrap confidence interval for $\mathbb{E}[Z]$ is given by:

$$U^\alpha = \bar{Z} + (z_\alpha^* - \bar{Z})$$

with $z_\alpha^*$ denoting the $\alpha$-quantile of the bootstrap estimates. It's worth noting that there exist multiple methods to compute bootstrap confidence intervals, including the basic, percentile, and bias-corrected approaches, and the method described above serves as a general illustration.

## C  SENSITIVITY ANALYSIS FOR $\Delta(h)$

Recall from Proposition 3.4 that the lower confidence intervals for $\tau_{\text{fair}}^*$ include a $\Delta(h)$ term which is defined as

$$\Delta(h) := \Phi_{\text{fair}}(h) - \tau_{\text{fair}}^*(\text{acc}(h)) \geq 0.$$

In other words, $\Delta(h)$ quantifies how 'far' the fairness loss of classifier $h$ (i.e. $\Phi_{\text{fair}}(h)$) is from the minimum attainable fairness loss for classifiers with accuracy $\text{acc}(h)$, (i.e. $\tau_{\text{fair}}^*(\text{acc}(h))$). This quantity is unknown in general and therefore, a practical strategy of obtaining lower confidence intervals on $\tau_{\text{fair}}^*(\psi)$ may involve positing values for $\Delta(h)$ which encode our belief on how close the fairness loss $\Phi_{\text{fair}}(h)$ is to $\tau_{\text{fair}}^*(\text{acc}(h))$. For example, when we assume that the classifier $h$ achieves the optimal accuracy-fairness tradeoff, i.e. $\Phi_{\text{fair}}(h) = \tau_{\text{fair}}^*(\text{acc}(h))$ then $\Delta(h) = 0$.

However, the assumption $\Phi_{\text{fair}}(h) = \tau_{\text{fair}}^*(\text{acc}(h))$ may not hold in general because we only have a finite training dataset and consequently the empirical loss minimisation may not yield the optima to the true expected loss. Moreover, the regularised loss used in training $h$ is a surrogate loss which approximates the solution to the constrained minimisation problem in Eq. (1). This means that optimising this regularised loss is not guaranteed to yield the optimal classifier which achieves the optimal fairness $\tau_{\text{fair}}^*(\text{acc}(h))$ even in the case when we have access to an infinitely large training dataset. Therefore, to incorporate any belief on the sub-optimality of the classifier $h$, we may consider conducting sensitivity analyses to plausibly quantify $\Delta(h)$.

Let $h_\theta : \mathcal{X} \times \Lambda \to \mathbb{R}$ be the YOTO model. One strategy for sensitivity analysis involves training multiple models $\mathcal{M} := \{h^{(1)}, h^{(2)}, \ldots, h^{(k)}\} \subseteq \mathcal{H}$ by optimising the regularised losses for few different choices of $\lambda$.

$$\mathcal{L}_\lambda(\theta) = \mathbb{E}[\mathcal{L}_{\text{CE}}(h_\theta(X), Y)] + \lambda \mathcal{L}_{\text{fair}}(h_\theta).$$

Importantly, we do not require covering the full range of $\lambda$ values when training separate models $\mathcal{M}$, and our methodology remains valid even when $\mathcal{M}$ is a single model. Next, let $h_\lambda^* \in \mathcal{M} \cup \{h_\theta(\cdot, \lambda)\}$ be such that

$$h_\lambda^* = \underset{h' \in \mathcal{M} \cup \{h_\theta(\cdot, \lambda)\}}{\arg\min} \widehat{\Phi_{\text{fair}}}(h') \quad \text{subject to} \quad \widehat{\text{acc}}(h') \geq \widehat{\text{acc}}(h_\theta(\cdot, \lambda)). \tag{9}$$

Here, $\widehat{\Phi_{\text{fair}}}$ and $\widehat{\text{acc}}$ denote the finite sample estimates of the fairness loss and model accuracy respectively. We treat the model $h_\lambda^*$ as a model which attains the optimum trade-off when estimating subject to the constraint $\text{acc}(h) \geq \text{acc}(h_\theta(\cdot, \lambda))$. Specifically, we use the empirical estimate $\widehat{\Delta}(h_\theta(\cdot, \lambda)) := \widehat{\Phi_{\text{fair}}}(h_\theta(\cdot, \lambda)) - \widehat{\Phi_{\text{fair}}}(h_\lambda^*) \geq 0$ as a plausible surrogate value for $\Delta(h_\theta(\cdot, \lambda))$, i.e., we posit

$$\Delta(h_\theta(\cdot, \lambda)) \leftarrow \widehat{\Delta}(h_\theta(\cdot, \lambda)) := \widehat{\Phi_{\text{fair}}}(h_\theta(\cdot, \lambda)) - \widehat{\Phi_{\text{fair}}}(h_\lambda^*).$$

Next, we can use this posited value of $\Delta(h_\theta(\cdot, \lambda))$ to construct the lower confidence interval using the following corollary of Proposition 3.4:

**Corollary C.1.** *Consider the YOTO model $h_\theta : \mathcal{X} \times \Lambda \to \mathbb{R}$. Given $\lambda_0 \in \Lambda$, let $U_{\text{acc}}^\alpha, L_{\text{fair}}^\alpha \in [0, 1]$ be such that*

$$\mathbb{P}(\text{acc}(h_\theta(\cdot, \lambda_0)) \leq U_{\text{acc}}^\alpha) \geq 1 - \alpha/2 \quad \text{and} \quad \mathbb{P}(\Phi_{\text{fair}}(h_\theta(\cdot, \lambda_0)) \geq L_{\text{fair}}^\alpha) \geq 1 - \alpha/2.$$

*Then, we have that $\mathbb{P}(\tau_{\text{fair}}^*(U_{\text{acc}}^\alpha) \geq L_{\text{fair}}^\alpha - \Delta(h_\theta(\cdot, \lambda_0))) \geq 1 - \alpha$.*

This result shows that if the goal is to construct lower confidence intervals on $\tau_{\text{fair}}^*(\psi)$ and we obtain that $\psi \geq U_{\text{acc}}^\alpha$, then using the monotonicity of $\tau_{\text{fair}}^*$ we have that $\tau_{\text{fair}}^*(\psi) \geq \tau_{\text{fair}}^*(U_{\text{acc}}^\alpha)$. Therefore the interval $[L_{\text{fair}}^\alpha - \Delta(h_\theta(\cdot, \lambda_0))), 1]$ serves as a lower confidence interval for $\tau_{\text{fair}}^*(\psi)$.

**When YOTO satisfies Pareto optimality, $\Delta(h_\theta(\cdot, \lambda)) \to 0$ as $|\mathcal{D}_{\textbf{cal}}| \to \infty$:** Here, we show that in the case when YOTO achieves the optimal trade-off, then our sensitivity analysis leads to $\Delta(h_\theta(\cdot, \lambda)) = 0$ as the calibration data size increases for all $\lambda \in \Lambda$. Our arguments in this section are not formal, however, this idea can be formalised without any significant difficulty.

First, the concept of Pareto optimality (defined below) formalises the idea that YOTO achieves the optimal trade-off:

**Assumption C.2** (Pareto optimality)**.**

If for some $\lambda \in \Lambda$ and $h' \in \mathcal{H}$ we have that, $\text{acc}(h') \geq \text{acc}(h_\theta(\cdot, \lambda))$ then, $\Phi_{\text{fair}}(h') \geq \Phi_{\text{fair}}(h_\theta(\cdot, \lambda))$,

In the case when YOTO satisfies this optimality property, then it is straightforward to see that $\Delta(h_\theta(\cdot, \lambda)) = 0$ for all $\lambda \in \Lambda$. In this case, as $\mathcal{D}_{\text{cal}} \to \infty$, we get that Eq. (9) roughly becomes

$$h_\lambda^* = \underset{h' \in \mathcal{M} \cup \{h_\theta(\cdot, \lambda)\}}{\arg\min} \Phi_{\text{fair}}(h') \quad \text{subject to} \quad \text{acc}(h') \geq \text{acc}(h_\theta(\cdot, \lambda)).$$

Here, Assumption C.2 implies that $h_\lambda^* = h_\theta(\cdot, \lambda)$, and therefore

$$\Delta(h_\theta(\cdot, \lambda)) \leftarrow \widehat{\Delta}(h_\theta(\cdot, \lambda)) := \widehat{\Phi_{\text{fair}}}(h_\theta(\cdot, \lambda)) - \widehat{\Phi_{\text{fair}}}(h_\lambda^*) = 0.$$

**Intuition behind our sensitivity analysis procedure** Intuitively, the high-level idea behind our sensitivity analysis is that it checks if we train models separately for fixed values of $\lambda$ (i.e. models in $\mathcal{M}$), how much better do these separately trained models perform in terms of the accuracy-fairness trade-offs as compared to our YOTO model. If we find that the separately trained models achieve a better trade-off than the YOTO model for specific values of $\lambda$, then the sensitivity analysis adjusts the empirical trade-off obtained using YOTO models (using the $\widehat{\Delta}(h_\theta(\cdot, \lambda))$ term defined above). If, on the other hand, we find that the YOTO model achieves a better trade-off than the separately trained models in $\mathcal{M}$, then the sensitivity analysis has no effect on the lower confidence intervals as in this case $\widehat{\Delta}(h_\theta(\cdot, \lambda)) = 0$.

## C.1 EXPERIMENTAL RESULTS

Here, we include empirical results showing how the CIs constructed change as a result of our sensitivity analysis procedure. In Figures 5 and 6, we include examples of CIs where the empirical trade-off obtained using YOTO is sub-optimal. In these cases, the lower CIs obtained without sensitivity analysis (i.e. when we assume $\Delta(h) = 0$) do not cover the empirical trade-offs for the separately trained models. However, the figures show that the sensitivity analysis procedure adjusts the lower CIs in both cases so that they encapsulate the empirical trade-offs that were not captured without sensitivity analysis.

Recall that $\mathcal{M}$ represents the set of additional separately trained models used for the sensitivity analysis. It can be seen from Figures 5 and 6 that in both cases our sensitivity analysis performs well with as little as two models (i.e. $|\mathcal{M}| = 2$), which shows that our sensitivity analysis does not come at a significant computational cost.

Additionally, in Figure 7 we also consider an example where YOTO achieves a better empirical trade-off than most other baselines considered, and therefore there is no need for sensitivity analysis. In this case, Figure 7 shows that sensitivity analysis has no effect on the CIs constructed since in this case sensitivity analysis gives us $\widehat{\Delta}(h_\theta(\cdot, \lambda)) = 0$ for $\lambda \in \Lambda$. This shows that in cases where sensitivity analysis is not needed (for example, if YOTO achieves optimal empirical trade-off), our sensitivity analysis procedure does not make the CIs more conservative.

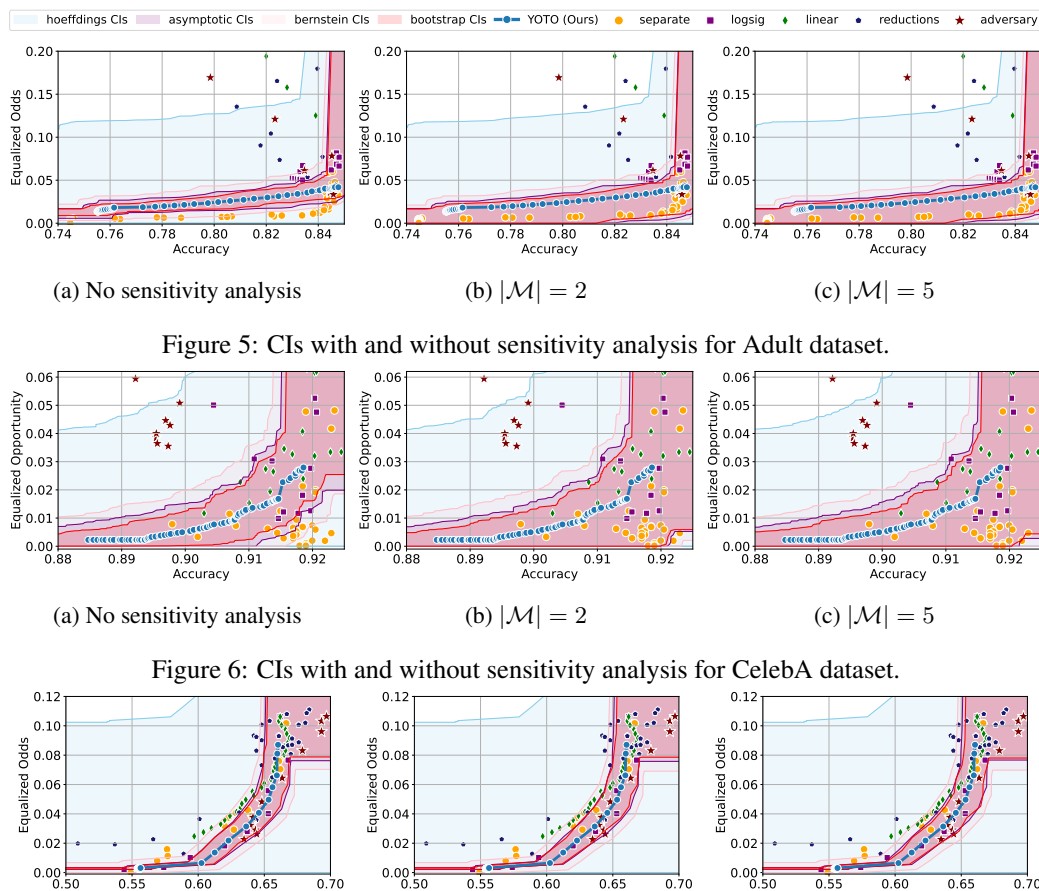

Figure 5: CIs with and without sensitivity analysis for Adult dataset.

Figure 6: CIs with and without sensitivity analysis for CelebA dataset.

Figure 7: CIs with and without sensitivity analysis for COMPAS dataset. Here, sensitivity analysis has no effect on the constructed CIs as the YOTO model achieves a better empirical trade-off than the separately trained models.

### C.2 LOWER BOUND FOR AN ARBITRARY $\Delta(h)$

Suppose the model that corresponds to $\tau^*_{\text{fair}}(\text{acc}(h))$ is $h^*$. Then $\tau^*_{\text{fair}}(\text{acc}(h)) = \Phi_{\text{fair}}(h^*)$. Suppose that $h$ is the empirically optimal model we could have obtained using the training data: $h = \arg\min_{h' \in \mathcal{H}} \widehat{\Phi_{\text{fair}}}(h')$.

$$\begin{aligned}
\Delta(h) =& \Phi_{\text{fair}}(h) - \Phi_{\text{fair}}(h^*) \\
=& \widehat{\Phi_{\text{fair}}(h)} - \widehat{\Phi_{\text{fair}}(h^*)} \\
& + \Phi_{\text{fair}}(h) - \widehat{\Phi_{\text{fair}}(h)} + \widehat{\Phi_{\text{fair}}(h^*)} - \Phi_{\text{fair}}(h^*) \\
\leq& 0 + 2 \max_{h' \in \mathcal{H}} |\Phi_{\text{fair}}(h') - \widehat{\Phi_{\text{fair}}(h')}|
\end{aligned}$$

In the above, the first inequality uses the fact that $h$ is the empirically optimal model and therefore $\widehat{\Phi_{\text{fair}}(h)} - \widehat{\Phi_{\text{fair}}(h^*)} \leq 0$.

Assuming $\widehat{\Phi_{\text{fair}}(h')}$ is additive that $\sum_{(x,y) \in \mathcal{D}_{\text{train}}} \epsilon_x \cdot \Phi_{\text{fair}}(h', x, y)$ Wang et al. (2022), we can then invoke the Rademacher bound on the maximal deviation between risks and empirical risks over a

hypothesis space $\mathcal{H}$ (Vapnik, 1999) we have

$$\max_{h' \in \mathcal{H}} |\Phi_{\text{fair}}(h') - \widehat{\Phi_{\text{fair}}(h')}| \le \mathcal{R}(\Phi_{\text{fair}} \circ \mathcal{H}) + \sqrt{\frac{\log 1/\delta}{|\mathcal{D}_{\text{train}}|}}$$

with probability at least $1 - \delta$, where

$$\mathcal{R}(\Phi_{\text{fair}} \circ \mathcal{H}) := \mathbb{E}_{X,Y,\epsilon} \left[ \sup_{h' \in \mathcal{H}} \frac{1}{|\mathcal{D}_{\text{train}}|} \sum_{(x,y) \in \mathcal{D}_{\text{train}}} \epsilon_x \cdot \Phi_{\text{fair}}(h', x, y) \right]$$

$\epsilon$s are independent random variables with $\mathbb{P}(\epsilon = +1) = \mathbb{P}(\epsilon = -1) = \frac{1}{2}$.

If $\Phi_{\text{fair}}$ is $L$-Lipschitz continuous, by the Lipschitz composition property of Rademacher averages we have

$$\mathcal{R}(\Phi_{\text{fair}} \circ \mathcal{H}) \le L \cdot \mathcal{R}(\mathcal{H})$$

where $\mathcal{R}(\mathcal{H})$ is the Rademacher complexity defined for $\mathcal{H}$ and can be further bounded using the VC-dimension of $\mathcal{H}$.

The above bound is valid when the model class $\mathcal{H}$ has a relatively small VC dimension $\text{VC}(\mathcal{H})$. For example, one of the Radermacher bound further bounds as

$$\mathcal{R}(\mathcal{H}) \le \sqrt{\frac{2\text{VC}(\mathcal{H}) \cdot \log \frac{e \cdot n}{\text{VC}(\mathcal{H})}}{n}}$$

For linear models, $\text{VC}(\mathcal{H}) = d$, where $d$ is the dimension of $X$.

## D  SCARCE SENSITIVE ATTRIBUTES

Our methodology of obtaining confidence intervals on $\Phi_{\text{fair}}$ assumes access to the sensitive attributes $A$ for all data points in the held-out dataset $\mathcal{D}$. However, in practice, we may only have access to $A$ for a small proportion of the data in $\mathcal{D}$. In this case, a naïve strategy would involve constructing confidence intervals using only the data for which $A$ is available. However, since such data is scarce, the confidence intervals constructed are very loose.

Suppose that we additionally have access to a predictive model $f_{\mathcal{A}}$ which predicts the sensitive attributes $A$ using the features $X$. In this case, another simple strategy would be to simply impute the missing values of $A$, with the values $\hat{A}$ predicted using $f_{\mathcal{A}}$. However, this will usually lead to a biased estimate of the fairness violation $\Phi_{\text{fair}}(h)$, and hence is not very reliable unless the model $f_{\mathcal{A}}$ is highly accurate. In this section, we show how to utilise the data with missing sensitive attributes to obtain tighter and more accurate confidence intervals on $\tau^*_{\text{fair}}(\psi)$.

Formally, we consider $\mathcal{D}_{\text{cal}} = \mathcal{D} \cup \tilde{\mathcal{D}}$ where $\mathcal{D}$ denotes a data subset of size $n$ that contains sensitive attributes (i.e. we observe $A$) and $\tilde{\mathcal{D}}$ denotes the data subset of size $N$ for which we do not observe the sensitive attributes $A$, and $N \gg n$. Additionally, for both datasets, we have predictions of the sensitive attributes made by a machine-learning algorithm $f_{\mathcal{A}} : \mathcal{X} \to \mathcal{A}$, where $f_{\mathcal{A}}(X) \approx A$. Concretely we have that $\mathcal{D} = \{(X_i, A_i, Y_i, f_{\mathcal{A}}(X_i))\}_{i=1}^n$ and $\tilde{\mathcal{D}} = \{(\tilde{X}_i, \tilde{Y}_i, f_{\mathcal{A}}(\tilde{X}_i))\}_{i=1}^N$

**High-level methodology**  Our methodology is inspired by prediction-powered inference (Angelopoulos et al., 2023) which builds confidence intervals on the expected outcome $\mathbb{E}[Y]$ using data for which the true outcome $Y$ is only available for a small proportion of the dataset. In our setting, however, it is the sensitive attribute $A$ that is missing for the majority of the data (and not the outcome $Y$).

For $h \in \mathcal{H}$, let $\Phi_{\text{fair}}(h)$ be a fairness violation (such as DP or EO), and let $\widetilde{\Phi_{\text{fair}}}(h)$ be the corresponding fairness violation computed on the data distribution where $A$ is replaced by the surrogate sensitive attribute $f_{\mathcal{A}}(X)$. For example, in the case of DP violation, $\Phi_{\text{fair}}(h)$ and $\widetilde{\Phi_{\text{fair}}}(h)$ denote:

$$\Phi_{\text{fair}}(h) = |\mathbb{P}(h(X) = 1 \mid A = 1) - \mathbb{P}(h(X) = 1 \mid A = 0)|,$$

$$\widetilde{\Phi_{\text{fair}}}(h) = |\mathbb{P}(h(X) = 1 \mid f_{\mathcal{A}}(X) = 1) - \mathbb{P}(h(X) = 1 \mid f_{\mathcal{A}}(X) = 0)|.$$

We next construct the confidence intervals on $\Phi_{\text{fair}}(h)$ using the following steps:

1. Using $\mathcal{D}$, we construct intervals $C_\epsilon(\alpha; h)$ on $\epsilon(h) := \Phi_{\text{fair}}(h) - \widetilde{\Phi_{\text{fair}}}(h)$ satisfying

$$\mathbb{P}(\epsilon(h) \in C_\epsilon(\alpha; h)) \geq 1 - \alpha. \tag{10}$$

   Even though the size of $\mathcal{D}$ is small, we choose a methodology which yields tight intervals for $\epsilon(h)$ when $f_{\mathcal{A}}(X_i) = A_i$ with a high probability.

2. Next, using the dataset $\tilde{\mathcal{D}}$, we construct intervals $\tilde{C}_{\text{f}}(\alpha; h)$ on $\widetilde{\Phi_{\text{fair}}}(h)$ satisfying

$$\mathbb{P}(\widetilde{\Phi_{\text{fair}}}(h) \in \tilde{C}_{\text{f}}(\alpha; h)) \geq 1 - \alpha. \tag{11}$$

   This interval will also be tight as the size of $\tilde{\mathcal{D}}$, $N \gg n$.

Finally, using the union bound idea we combine the two confidence intervals to obtain the confidence interval for $\Phi_{\text{fair}}(h) - \widetilde{\Phi_{\text{fair}}}(h) + \widetilde{\Phi_{\text{fair}}}(h) = \Phi_{\text{fair}}(h)$. We make this precise in the following result:

**Lemma D.1.** *Let $C_\epsilon(\alpha; h), \tilde{C}_{\text{f}}(\alpha; h)$ be as defined in equations 10 and 11. Then, if we define $C_{\text{fair}}^\alpha(h) = \{x + y \,|\, x \in C_\epsilon(\alpha; h), y \in \tilde{C}_{\text{f}}(\alpha; h)\}$, we have that*

$$\mathbb{P}(\Phi_{\text{fair}}(h) \in C_{\text{fair}}^\alpha(h)) \geq 1 - 2\alpha.$$

When constructing the CIs over $\widetilde{\Phi_{\text{fair}}}(h)$ using imputed sensitive attributes $f_{\mathcal{A}}(X)$ in step 2 above, the prediction error of $f_{\mathcal{A}}$ introduces an error in the obtained CIs (denoted by $\epsilon(h)$). Step 1 rectifies this by constructing a CI over the incurred error $\epsilon(h)$, and therefore combining the two allows us to obtain intervals which utilise all of the available data while ensuring that the constructed CIs are well-calibrated.

**Example: Demographic parity**   Having defined our high-level methodology above, we concretely demonstrate how this can be applied to the case where the fairness loss under consideration is DP. As described above, the first step involves constructing intervals on $\epsilon(h) := \Phi_{\text{fair}}(h) - \widetilde{\Phi_{\text{fair}}}(h)$ using a methodology which yields tight intervals when $f_{\mathcal{A}}(X_i) = A_i$ with a high probability. To this end, we use bootstrapping as described in Algorithm 1.

Even though bootstrapping does not provide us with finite sample coverage guarantees, it is asymptotically exact and satisfies the property that the confidence intervals are tight when $\hat{A} = A$ with a high probability. On the other hand, concentration inequalities (such as Hoeffding's inequality) seek to construct confidence intervals individually on $\Phi_{\text{fair}}(h)$ and $\widetilde{\Phi_{\text{fair}}}(h)$ and subsequently combine them through union bounds argument, for example. In doing so, these methods do not account for how close the values of $\Phi_{\text{fair}}(h)$ and $\widetilde{\Phi_{\text{fair}}}(h)$ might be in the data.

To make this concrete, consider the example where $f_{\mathcal{A}}(X) \overset{\text{a.s.}}{=} A$ and hence $\Phi_{\text{fair}}(h) = \widetilde{\Phi_{\text{fair}}}(h)$. When using concentration inequalities to construct the $1 - \alpha$ confidence intervals on $\Phi_{\text{fair}}(h)$ and $\widetilde{\Phi_{\text{fair}}}(h)$, we obtain identical intervals for the two quantities, say $[l, u]$. Then, using union bounds we obtain that $\Phi_{\text{fair}}(h) - \widetilde{\Phi_{\text{fair}}}(h) \in [l - u, u - l]$ with probability at least $1 - 2\alpha$. In this case even though $\Phi_{\text{fair}}(h) - \widetilde{\Phi_{\text{fair}}}(h) = 0$, the width of the interval $[l - u, u - l]$ does not depend on the closeness of $\Phi_{\text{fair}}(h)$ and $\widetilde{\Phi_{\text{fair}}}(h)$ and therefore is not tight. Bootstrapping helps us circumvent this problem, since in this case for each resample of the data $\mathcal{D}$, the finite sample estimates $\widehat{\widetilde{\Phi_{\text{fair}}}(h)}$ and $\widehat{\widetilde{\Phi_{\text{fair}}}(h)}$ will be equal. We outline the bootstrapping algorithm below.

Using Algorithm 1 we construct a confidence interval $C_\epsilon(\alpha; h)$ on $\epsilon(h)$ of size $1 - \alpha$, which approximately satisfies Eq. (10). Next, using standard techniques we can obtain an interval $\tilde{C}_{\text{f}}(\alpha; h)$ on $\widetilde{\Phi_{\text{fair}}}(h)$ using $\tilde{\mathcal{D}}$ which satisfies Eq. (11). Like before, the interval $\tilde{C}_{\text{f}}(\alpha; h)$ is likely to be tight as we use $\tilde{\mathcal{D}}$ to construct it, which is significantly larger than $\mathcal{D}$. Finally, combining the two as shown in Lemma D.1, we obtain the confidence interval on $\Phi_{\text{fair}}(h)$.

## D.1   EXPERIMENTAL RESULTS

Here, we present experimental results in the setting where the sensitive attributes are missing for majority of the calibration data. Figures 8-13 show the results for different datasets and predictive

---

**Algorithm 1:** Bootstrapping for estimating $\epsilon(h) := \Phi_{\text{fair}}(h) - \widetilde{\Phi_{\text{fair}}}(h)$

---

**Input:** Dataset $\mathcal{D}$, number of bootstrap samples $B$, significance level $\alpha$
**Output:** $1 - \alpha$ confidence interval for $\epsilon(h)$
Initialize empty array $\mathbf{v}_b$
**for** $i \leftarrow 1$ **to** $B$ **do**

    Draw a bootstrap sample $\mathcal{D}^*$ of size $|\mathcal{D}|$ with replacement from $\mathcal{D}$

    Compute $\widehat{\Phi_{\text{fair}}(h)}$ and $\widehat{\widetilde{\Phi_{\text{fair}}}(h)}$ on $\mathcal{D}^*$

    Compute the difference $\widehat{\epsilon(h)} := \widehat{\Phi_{\text{fair}}(h)} - \widehat{\widetilde{\Phi_{\text{fair}}}(h)}$

    Append $\widehat{\epsilon(h)}$ to $\mathbf{v}_b$

Compute the $\alpha/2$ and $1 - \alpha/2$ quantiles of $\mathbf{v}_b$, denoted as $\tau^*_{\text{fair}}$ and $u$
**return** Confidence interval $C_\epsilon(\alpha; h) = [l, u]$.

---

models $f_{\mathcal{A}}$ with varying accuracies. Here, the empirical fairness violation values for both YOTO and separately trained models are evaluated using the true sensitive attributes over the entire calibration data.

**CIs with imputed sensitive attributes are mis-calibrated** Figures 8, 10 and 12 show results for Adult, COMPAS and CelebA datasets, where the CIs are computed by imputing the missing sensitive attributes with the predicted sensitive attributes $f_{\mathcal{A}}(X) \approx A$. The figures show that when the accuracy of $f_{\mathcal{A}}$ is below 90%, the CIs are highly miscalibrated as they do not entirely contain the empirical trade-offs for both YOTO and separately trained models.

**Our methodology corrects for the mis-calibration** In contrast, Figures 9, 11 and 13 which include the corresponding results using our methodology, show that our methodology is able to correct for the mis-calibration in CIs arising from the prediction error in $f_{\mathcal{A}}$. Even though the CIs obtained using our methodology are more conservative than those obtained by imputing the missing sensitive attributes with $f_{\mathcal{A}}(X)$, they are more well-calibrated and contain the empirical trade-offs for both YOTO and separately trained model.

**Imputing missing sensitive attributes may work when $f_{\mathcal{A}}$ has high accuracy** Finally, Figures 8c, 10c and 12c show that the CIs with imputed sensitive attributes are relatively better calibrated as the accuracy of $f_{\mathcal{A}}$ increases to 90%. In this case, the CIs with imputed sensitive attributes mostly contain empirical trade-offs. This shows that in cases where the predictive model $f_{\mathcal{A}}$ has high accuracy, it may be sufficient to impute missing sensitive attributes with $f_{\mathcal{A}}(X)$ when constructing the CIs.

## E    EXPERIMENTAL DETAILS AND ADDITIONAL RESULTS

In this section, we provide greater details regarding our experimental setup and models used. We first begin by defining the Equalized Odds metric which has been used in our experiments, along with DP and EOP.

**Equalized Odds (EO):** EO condition states that, both the true positive rates and false positive rates for all sensitive groups are equal, i.e. $\mathbb{P}(h(X) = 1 \mid A = a, Y = y) = \mathbb{P}(h(X) = 1 \mid Y = y)$ for any $a \in \mathcal{A}$ and $y \in \{0, 1\}$. The absolute EO violation is defined as:

$$\Phi_{\text{EO}}(h) := 1/2 \left| \mathbb{P}(h(X) = 1 \mid A = 1, Y = 1) - \mathbb{P}(h(X) = 1 \mid A = 0, Y = 1) \right|$$
$$+ 1/2 \left| \mathbb{P}(h(X) = 1 \mid A = 1, Y = 0) - \mathbb{P}(h(X) = 1 \mid A = 0, Y = 0) \right|.$$

Next, we provide additional details regarding the YOTO model.

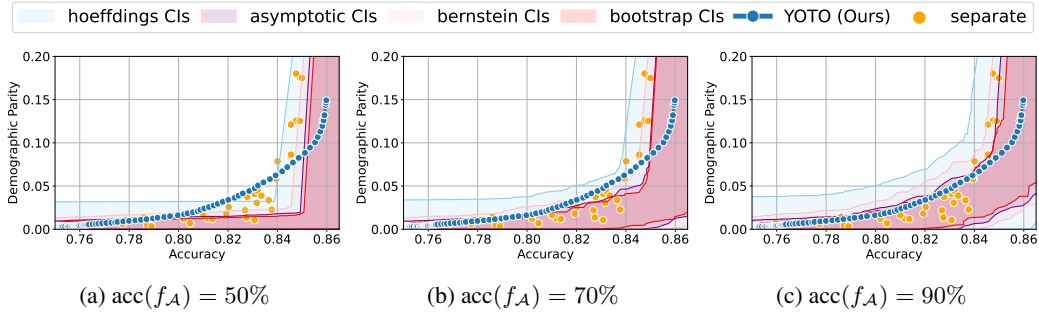

Figure 8: CIs obtained by imputing missing senstive attributes using $f_\mathcal{A}$ for Adult dataset. Here $n = 50$ and $N = 2500$.

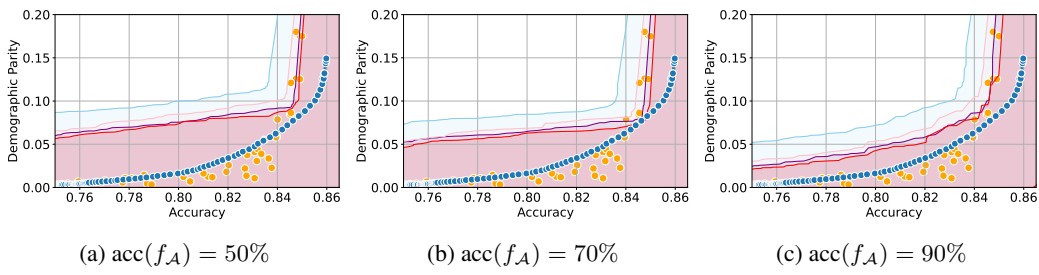

Figure 9: CIs were obtained using our methodology for the Adult dataset. Here $n = 50$ and $N = 2500$.

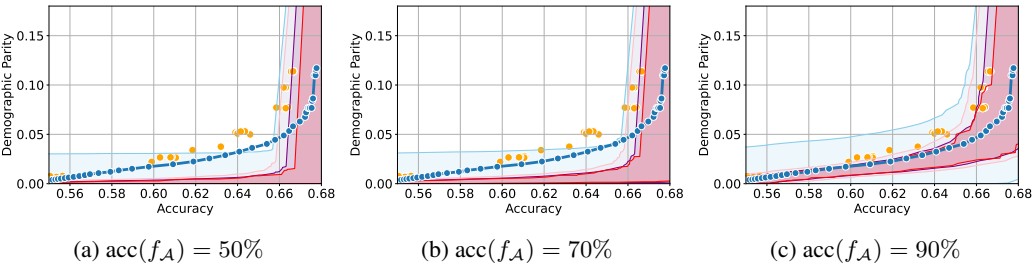

Figure 10: CIs obtained by imputing missing senstive attributes using $f_\mathcal{A}$ for COMPAS dataset. Here $n = 50$ and $N = 2000$.

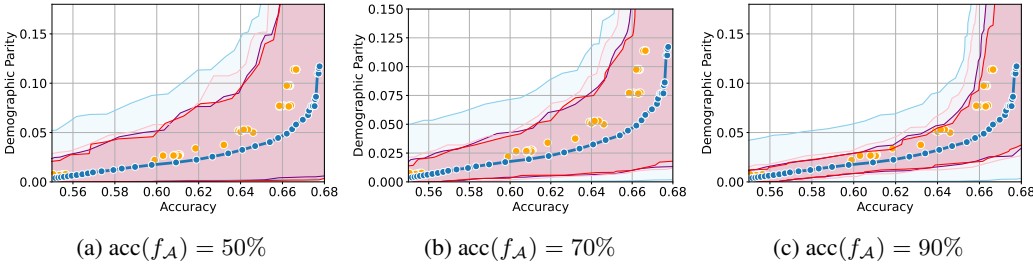

Figure 11: CIs obtained using our methodology for COMPAS dataset. Here $n = 50$ and $N = 2000$.

### E.1 PRACTICAL DETAILS REGARDING YOTO MODEL

As described in Section 3, we consider optimising regularized losses of the form

$$\mathcal{L}_\lambda(\theta) = \mathbb{E}[\mathcal{L}_{\mathrm{CE}}(h_\theta(X), Y)] + \lambda\, \mathcal{L}_{\mathrm{fair}}(h_\theta).$$

When training YOTO models, instead of fixing $\lambda$, we sample the parameter $\lambda$ from a distribution $P_\lambda$. As a result, during training the model observes many different values of $\lambda$ and learns to optimise the loss $\mathcal{L}_\lambda$ for all of them simultaneously. At inference time, the model can be conditioned on a chosen

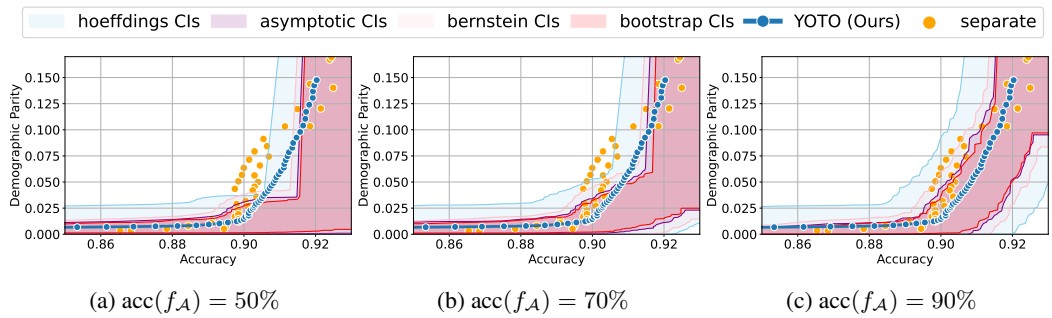

(a) $\text{acc}(f_{\mathcal{A}}) = 50\%$   (b) $\text{acc}(f_{\mathcal{A}}) = 70\%$   (c) $\text{acc}(f_{\mathcal{A}}) = 90\%$

Figure 12: CIs obtained by imputing missing senstive attributes using $f_{\mathcal{A}}$ for CelebA dataset. Here $n = 50$ and $N = 2500$.

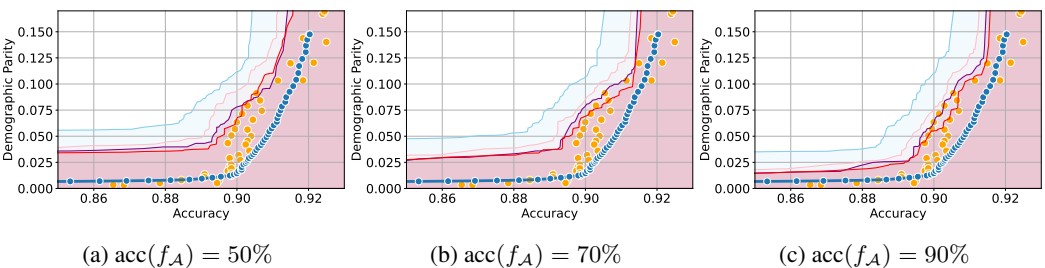

(a) $\text{acc}(f_{\mathcal{A}}) = 50\%$   (b) $\text{acc}(f_{\mathcal{A}}) = 70\%$   (c) $\text{acc}(f_{\mathcal{A}}) = 90\%$

Figure 13: CIs obtained using our methodology for CelebA dataset. Here $n = 50$ and $N = 2500$.

parameter value $\lambda'$ and recovers the model trained to optimise $\mathcal{L}_{\lambda'}$. The loss being minimised can thus be expressed as follows:

$$\underset{h_\theta : \mathcal{X} \times \Lambda \to \mathbb{R}}{\arg\min} \; \mathbb{E}_{\lambda \sim P_\lambda} \left[ \mathbb{E}[\mathcal{L}_{\text{CE}}(h_\theta(X, \lambda), Y)] + \lambda \, \mathcal{L}_{\text{fair}}(h_\theta(\cdot, \lambda)) \right].$$

The fairness losses $\mathcal{L}_{\text{fair}}$ considered for the YOTO model are:

DP: $\quad \mathcal{L}_{\text{fair}}(h_\theta(\cdot, \lambda)) = |\mathbb{E}[\sigma(h_\theta(X, \lambda)) \mid A = 1] - \mathbb{E}[\sigma(h_\theta(X, \lambda)) \mid A = 0]|$

EOP: $\quad \mathcal{L}_{\text{fair}}(h_\theta(\cdot, \lambda)) = |\mathbb{E}[\sigma(h_\theta(X, \lambda)) \mid A = 1, Y = 1] - \mathbb{E}[\sigma(h_\theta(X, \lambda)) \mid A = 0, Y = 1]|$

EO: $\quad \mathcal{L}_{\text{fair}}(h_\theta(\cdot, \lambda)) = |\mathbb{E}[\sigma(h_\theta(X, \lambda)) \mid A = 1, Y = 1] - \mathbb{E}[\sigma(h_\theta(X, \lambda)) \mid A = 0, Y = 1]|$
$\quad\quad + |\mathbb{E}[\sigma(h_\theta(X, \lambda)) \mid A = 1, Y = 0] - \mathbb{E}[\sigma(h_\theta(X, \lambda)) \mid A = 0, Y = 0]|.$

Here, $\sigma(x) := 1/(1 + e^{-x})$ denotes the sigmoid function.

In our experiments, we sample a new $\lambda$ for every batch. Moreover, we use the log-uniform distribution as per Dosovitskiy & Djolonga (2020) as the sampling distribution $P_\lambda$, where the uniform distribution is $U[10^{-6}, 10]$. To condition the network on $\lambda$ parameters, we follow in the footsteps of Dosovitskiy & Djolonga (2020) to use Feature-wise Linear Modulation (FiLM) (Perez et al., 2017). For completeness, we include the description of the architecture next.

Initially, we determine which network layers should be conditioned, which can encompass all layers or just a subset. For each chosen layer, we condition it based on the weight parameters $\lambda$. Given a layer that yields a feature map $f$ with dimensions $W \times H \times C$, where $W$ and $H$ denote the spatial dimensions and $C$ stands for the channels, we introduce the parameter vector $\lambda$ to two distinct multi-layer perceptrons (MLPs), denoted as $M_\sigma$ and $M_\mu$. These MLPs produce two vectors, $\sigma$ and $\mu$, each having a dimensionality of $C$. The feature map is then transformed by multiplying it channel-wise with $\sigma$ and subsequently adding $\mu$. The resultant transformed feature map $f'$ is given by:

$$f'_{ijk} = \sigma_k f_{ijk} + \mu_k \quad \text{where} \quad \sigma = M_\sigma(\lambda) \quad \text{and} \quad \mu = M_\mu(\lambda).$$

Next, we provide exact architectures we used for each dataset in our experiments.

### E.1.1 YOTO ARCHITECTURES

**Adult and COMPAS dataset** Here, we use a simple logistic regression as the main model, with only the scalar logit outputs of the logistic regression being conditioned using FiLM. The MLPs

$M_\mu, M_\sigma$ both have two hidden layers, each of size 4, and ReLU activations. We train the model for a maximum of 1000 epochs, with early stopping based on validation losses. Training these simple models takes roughly 5 minutes on a Tesla-V100-SXM2-32GB GPU.

**CelebA dataset**    For the CelebA dataset, our architecture is a convolutional neural network (ConvNet) integrated with the FiLM (Feature-wise Linear Modulation) mechanism. The network starts with two convolutional layers: the first layer has 32 filters with a kernel size of $3 \times 3$, and the second layer has 64 filters, also with a $3 \times 3$ kernel. Both convolutional layers employ a stride of 1 and are followed by a max-pooling layer that reduces each dimension by half.

The feature maps from the convolutional layers are flattened and passed through a series of fully connected (MLP) layers. Specifically, the first layer maps the features to 64 dimensions, and the subsequent layers maintain this size until the final layer, which outputs a scalar value. The activation function used in these layers is ReLU.

To condition the network on the $\lambda$ parameter using FiLM, we design two multi-layer perceptrons (MLPs), $M_\mu$ and $M_\sigma$. Both MLPs take the $\lambda$ parameter as input and have 4 hidden layers. Each of these hidden layers is of size 256. These MLPs produce the modulation parameters $\mu$ and $\sigma$, which are used to perform feature-wise linear modulation on the outputs of the main MLP layers. The final output of the network is passed through a sigmoid activation function to produce the model's prediction. We train the model for a maximum of 1000 epochs, with early stopping based on validation losses. Training this model takes roughly 1.5 hours on a Tesla-V100-SXM2-32GB GPU.

**Jigsaw dataset**    For the Jigsaw dataset, we employ a neural network model built upon the BERT architecture (Devlin et al., 2018) integrated with the Feature-wise Linear Modulation (FiLM) mechanism. We utilize the representation corresponding to the [CLS] token, which carries aggregate information about the entire sequence. To condition the BERT's output on the $\lambda$ parameter using FiLM, we design two linear layers, which map the $\lambda$ parameter to modulation parameters $\gamma$ and $\beta$, both of dimension equal to BERT's hidden size of 768. These modulation parameters are then used to perform feature-wise linear modulation on the [CLS] representation. The modulated representation is passed through a classification head, which consists of a linear layer mapping from BERT's hidden size (768) to a scalar output. In terms of training details, our model is trained for a maximum of 10 epochs, with early stopping based on validation losses. Training this model takes roughly 6 hours on a Tesla-V100-SXM2-32GB GPU.

### E.2    DATASETS

We used four real-world datasets for our experiments.

**Adult dataset**    The Adult income dataset (Becker & Kohavi, 1996) includes employment data for 48,842 individuals where the task is to predict whether an individual earns more than \$50k per year and includes demographic attributes such as age, race and gender. In our experiments, we consider gender as the sensitive attribute.

**COMPAS dataset**    The COMPAS recidivism data comprises collected by ProPublica (Angwin et al., 2016), includes information for 6172 defendants from Broward County, Florida. This information comprises 52 features including defendants' criminal history and demographic attributes, and the task is to predict recidivism for defendants. The sensitive attribute in this dataset is the defendants' race where $A = 1$ represents 'African American' and $A = 0$ corresponds to all other races.

**CelebA dataset**    The CelebA dataset (Liu et al., 2015) consists of 202,599 celebrity images annotated with 40 attribute labels. In our task, the objective is to predict whether an individual in the image is smiling. The dataset comprises features in the form of image pixels and additional attributes such as hairstyle, eyeglasses, and more. The sensitive attribute for our experiments is gender.

**Jigsaw Toxicity Classification dataset**    The Jigsaw Toxicity Classification dataset (Jigsaw & Google, 2019) contains online comments from various platforms, aimed at identifying and miti-

gating toxic behavior online. The task is to predict whether a given comment is toxic or not. The dataset includes features such as the text of the comment and certain metadata such as the gender or race to which each comment relates. In our experiments, the sensitive attribute is the gender to which the comment refers, and we only filter the comments which refer to exactly one of 'male' or 'female' gender. This leaves us with 107,106 distinct comments.

### E.3 BASELINES

The baselines considered in our experiments include:

- **Regularization based approaches (Bendekgey & Sudderth, 2021):** These methods seek to minimise fairness loss using regularized losses as shown in Section 3. We consider different regularization terms $\mathcal{L}_{\text{fair}}(h_\theta)$ as smooth relaxations of the fairness violation $\Phi_{\text{fair}}$ as proposed in the literature (Bendekgey & Sudderth, 2021). To make this concrete, when the fairness violation under consideration is DP, we consider

$$\mathcal{L}_{\text{fair}}(h_\theta) = \mathbb{E}[g(h_\theta(X)) \mid A = 1] - \mathbb{E}[g(h_\theta(X)) \mid A = 0],$$

  with $g(x) = x$ denoted as 'linear' in our results and $g(x) = \log \sigma(x)$ where $\sigma(x) := 1/(1 + e^{-x})$, denoted as 'logsig' in our results. In addition to these methods, we also consider separately trained models with the same regularization term as the YOTO models, i.e.,

  DP: $\quad \mathcal{L}_{\text{fair}}(h_\theta) = |\mathbb{E}[\sigma(h_\theta(X)) \mid A = 1] - \mathbb{E}[\sigma(h_\theta(X)) \mid A = 0]|$

  EOP: $\quad \mathcal{L}_{\text{fair}}(h_\theta) = |\mathbb{E}[\sigma(h_\theta(X)) \mid A = 1, Y = 1] - \mathbb{E}[\sigma(h_\theta(X)) \mid A = 0, Y = 1]|$

  EO: $\quad \mathcal{L}_{\text{fair}}(h_\theta) = |\mathbb{E}[\sigma(h_\theta(X)) \mid A = 1, Y = 1] - \mathbb{E}[\sigma(h_\theta(X)) \mid A = 0, Y = 1]|$
  $\quad\quad + |\mathbb{E}[\sigma(h_\theta(X)) \mid A = 1, Y = 0] - \mathbb{E}[\sigma(h_\theta(X)) \mid A = 0, Y = 0]|.$

  Here, $\sigma(x) := 1/(1 + e^{-x})$ denotes the sigmoid function. We denote these models as 'separate' in our experimental results as they are the separately trained counterparts to the YOTO model. For each relaxation, we train models for a range of $\lambda$ values uniformly chosen in $[0, 10]$ interval.

- **Reductions Approach (Agarwal et al., 2018):** This method transforms the fairness problem into a sequence of cost-sensitive classification problems. Like the regularization approaches this requires multiple models to be trained. Here, to try and reproduce the trade-off curves, we train the reductions approach with a range of different fairness constraints uniformly in $[0, 1]$.

- **Adversarial Approaches (Zhang et al., 2018):** These methods utilize an adversarial training paradigm where an additional model, termed the adversary, is introduced during training. The primary objective of this adversary is to predict the sensitive attribute $A$ using the predictions $h_\theta(X)$ generated by the main classifier $h_\theta$. The training process involves an adversarial game between the primary classifier and the adversary, striving to achieve equilibrium. This adversarial dynamic ensures that the primary classifier's predictions are difficult to use for determining the sensitive attribute $A$, thereby minimizing unfair biases associated with $A$. Specifically, for DP constraints, the adversary takes the logit outputs of the classifier as the input and predicts $A$. In contrast for EO and EOP constraints, the adversary also takes the true label $Y$ as the input. For EOP constraint, the adversary is only trained on data with $Y = 1$.

### E.4 ADDITIONAL RESULTS

In Figures 14-25 we include additional results for all datasets and fairness violations with an increasing number of calibration data $\mathcal{D}_{\text{cal}}$. It can be seen that as the number of calibration data increases, the CIs constructed become increasingly tighter. However, the asymptotic, Bernstein and bootstrap CIs are informative even when the calibration data is as little as 500 for COMPAS data (Figures 17-19) and 1000 for all other datasets. These results show that the larger the calibration data $\mathcal{D}_{\text{cal}}$, the tighter the constructed CIs are likely to be. However, even in cases where the calibration dataset is relatively small, we obtain informative CIs in most cases.

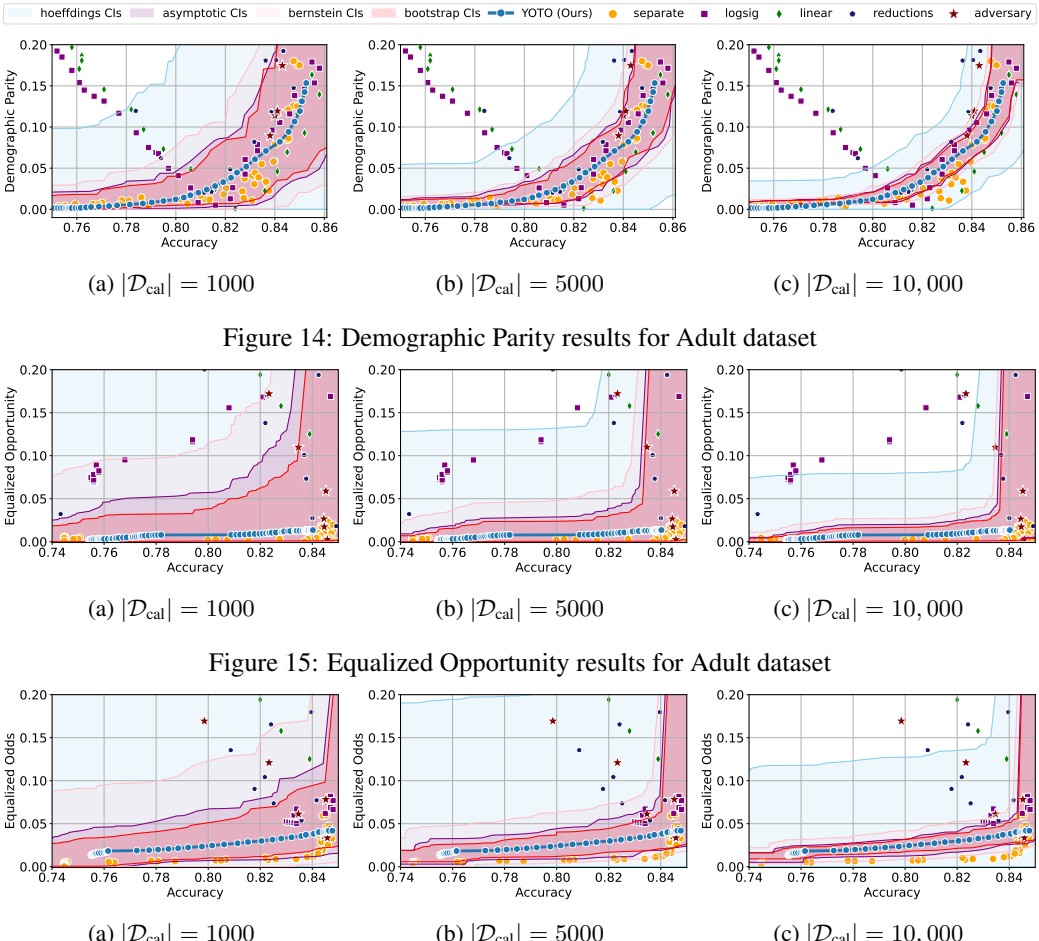

Figure 14: Demographic Parity results for Adult dataset

Figure 15: Equalized Opportunity results for Adult dataset

Figure 16: Equalized Odds results for Adult dataset

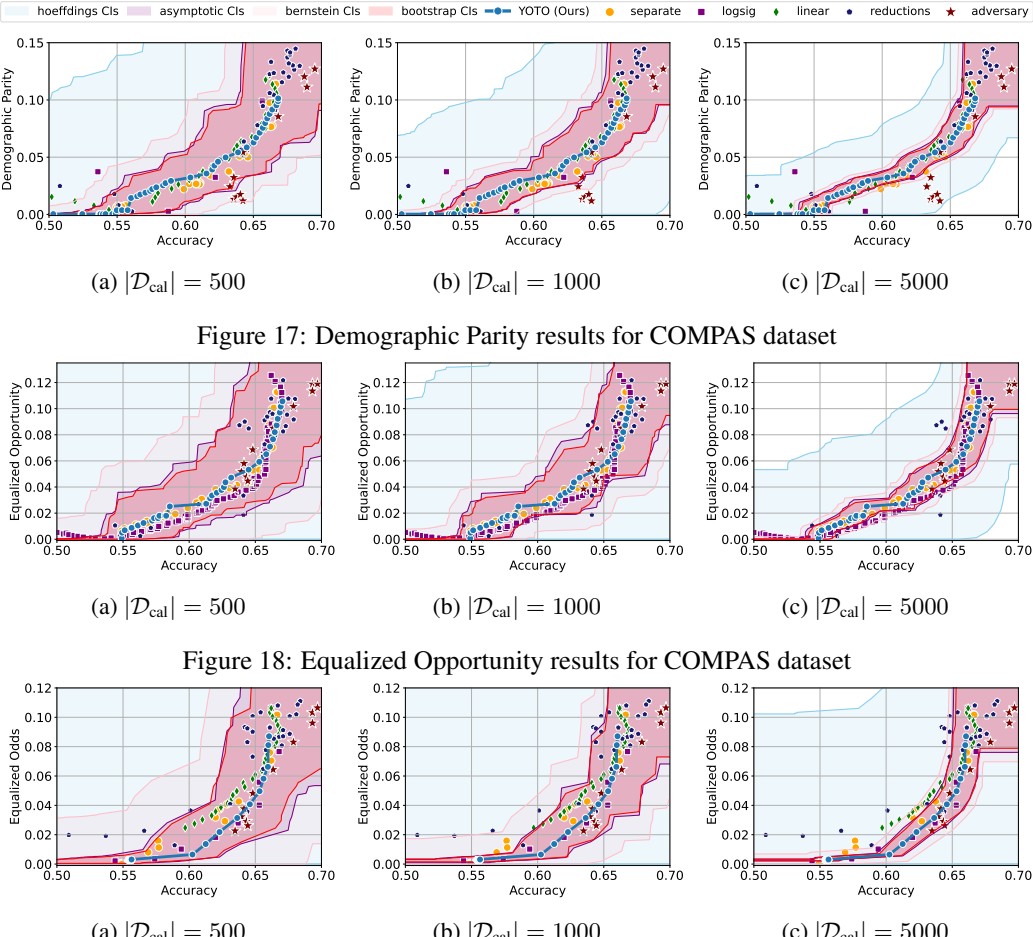

Figure 17: Demographic Parity results for COMPAS dataset

Figure 18: Equalized Opportunity results for COMPAS dataset

Figure 19: Equalized Odds results for COMPAS dataset

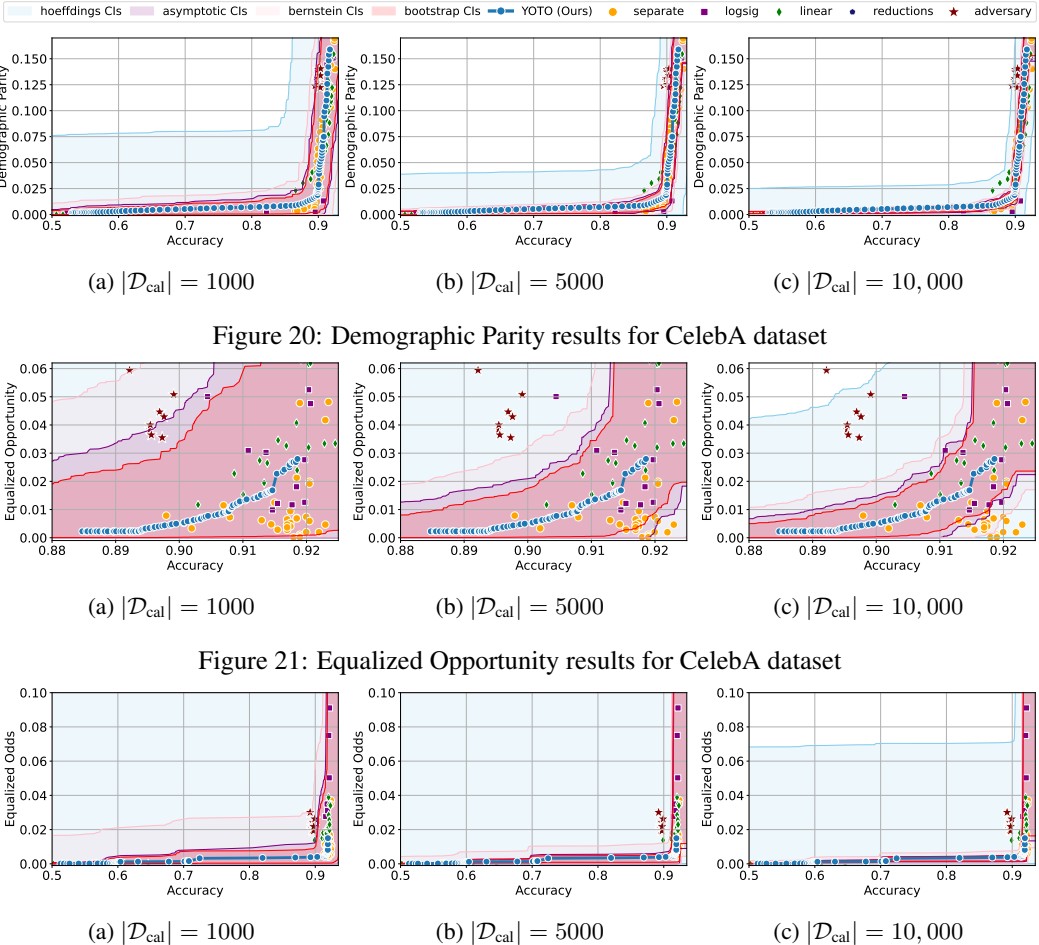

Figure 20: Demographic Parity results for CelebA dataset

Figure 21: Equalized Opportunity results for CelebA dataset

Figure 22: Equalized Odds results for CelebA dataset

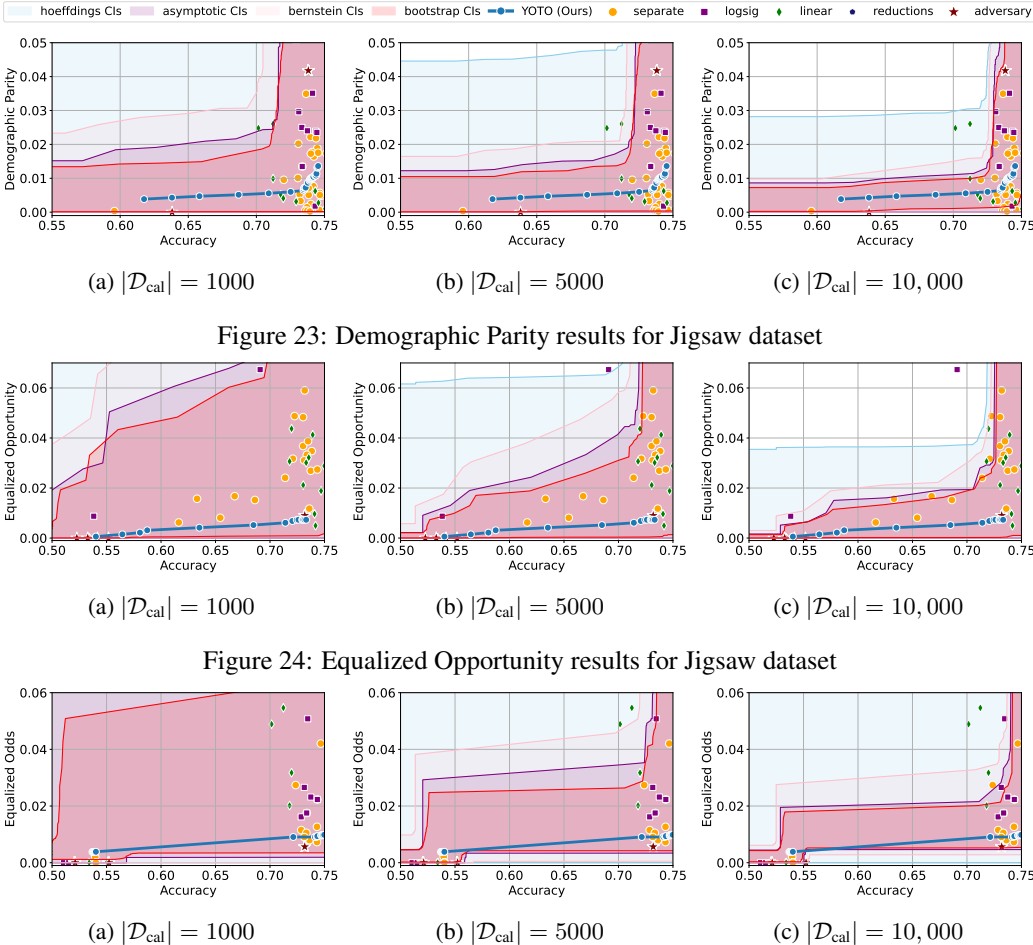

Figure 23: Demographic Parity results for Jigsaw dataset

Figure 24: Equalized Opportunity results for Jigsaw dataset

Figure 25: Equalized Odds results for Jigsaw dataset

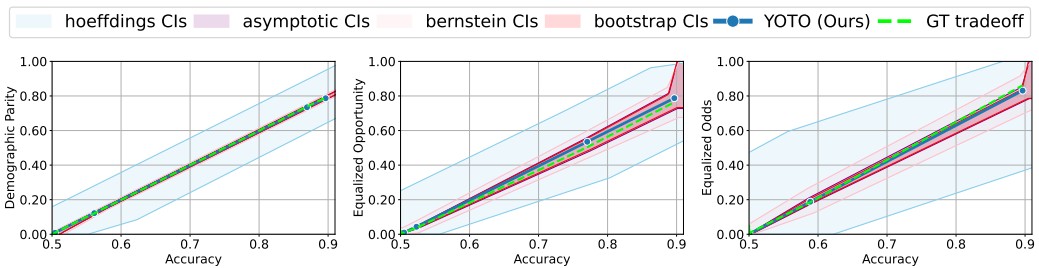

Figure 26: Results for the synthetic dataset with ground truth trade-off curves $\tau^*_{\text{fair}}$.

## E.5 SYNTHETIC DATA EXPERIMENTS

In real-world settings, the ground truth trade-off curve $\tau^*_{\text{fair}}$ remains intractable because we only have access to a finite dataset. In this section, we consider a synthetic data setting, where the ground truth trade-off curve can be obtained, to verify that the YOTO trade-off curves are consistent with the ground truth and that the confidence intervals obtained using our methodology contain $\tau^*_{\text{fair}}$.

**Dataset** Here, we consider a setup with $\mathcal{X} = \mathbb{R}$, $\mathcal{A} = \{0, 1\}$ and $\mathcal{Y} = \{0, 1\}$. Specifically, $A \sim \text{Bern}(0.5)$ and we define the conditional distributions $X \mid A = a$ as:

$$X \mid A = a \sim \mathcal{N}(a, 0.2^2)$$

Moreover, we define the labels $Y$ as follows:

$$Y = Z \, \mathbb{1}(X > 0.5) + (1 - Z) \, \mathbb{1}(X \leq 0.5),$$

where $Z \sim \text{Bern}(0.9)$ and $Z \perp\!\!\!\perp X$. Here, $Z$ introduces some 'noise' to the labels $Y$ and means that perfect accuracy is not achievable by linear classifiers. If perfect accuracy was achievable, the optimal values for Equalized Odds and Equalized Opportunity would be 0 (and would be achieved by the perfect classifier), therefore our use of 'noisy' labels $Y$ ensures that the ground truth trade-off curves will be non-trivial.

**YOTO model training** Using the data generating We generate 5000 training datapoints, which we use to train the YOTO model. The YOTO model for this dataset comprises of a simple logistic regression as the main model, with only the scalar logit outputs of the logistic regression being conditioned using FiLM. The MLPs $M_\mu, M_\sigma$ both have two hidden layers, each of size 4, and ReLU activations. We train the model for a maximum of 1000 epochs, with early stopping based on validation losses. Training these simple model only requires one CPU and takes roughly 2 minutes.

**Ground truth trade-off curve** To obtain the ground truth trade-off curve $\tau^*_{\text{fair}}$, we consider the family of classifiers

$$h_c(X) = \mathbb{1}(X > c)$$

for $c \in \mathbb{R}$. Next, we calculate the trade-offs achieved by this model family for a fine grid of $c$ values between -3 and 3, using a dataset of size 500,000 obtained using the data-generating mechanism described above. The large dataset size ensures that the finite sample errors in accuracy and fairness violation values are negligible. This allows us to reliably plot the trade-off curve $\tau^*_{\text{fair}}$.

### E.5.1 RESULTS

Figure 26 shows the results for the synthetic data setup for three different fairness violations, obtained using a calibration dataset $\mathcal{D}_{\text{cal}}$ of size 2000. It can be seen that for each fairness violation considered, the YOTO trade-off curve aligns very well with the ground-truth trade-off curve $\tau^*_{\text{fair}}$. Additionally, we also consider four different confidence intervals obtained using our methodology, and Figure 26 shows that each of the four confidence intervals considered contain the ground-truth trade-off curve. This empirically verifies the validity of our confidence intervals in this synthetic setting.

