# OpenReview forum: "Dataset Fairness: Achievable Fairness On Your Data With Utility Guarantees"
_ICLR.cc/2024/Conference — Submitted to ICLR 2024_

### Official Review · Reviewer_PBf8 · 2023-11-05

**Soundness:** 2 fair
**Presentation:** 3 good
**Contribution:** 2 fair
**Rating:** 6
**Confidence:** 2

**Summary:**

This paper proposes a method to estimate the accuracy-fairness trade-off curve given the dataset and model class. It first uses an existing method, You-Only-Train-Once (YOTO), to get the trade-off curve efficiently without training multiple models. It then proposes a way to obtain the confidence intervals.

**Strengths:**

The accuracy-fairness curve is widely used in the algorithmic fairness community. How to set a range of achievable fairness violations is a practical problem. The method proposed in this paper provides us with an efficient solution for estimating the curve with two-sided confidence intervals.

**Weaknesses:**

1. The method is highly based on an existing method, YOTO. Although it doesn't need to train multiple models, the cost of YOTO is not discussed in this paper.
2. The accuracy-fairness curve is algorithm-agnostic. Figure 4 shows that the estimated curve has more errors when using some particular algorithmic fairness methods. However, people tend to use some algorithmic fairness methods to train the model. In that case, the practical use of the curve is concerned.
3. The method requires an in-distribution calibration dataset and only applies to in-distribution tests.

**Questions:**

1. What is the L_fair in equation (2)?
2. What are the costs and limitations of YOTO?
3. Can we extend the method to be algorithm-sensitive? For example, suppose we know what algorithmic fairness method to use, can we estimate the curve for that algorithm?

---

> ### Author Response · Authors · 2023-11-17
> **Rebuttal by Authors**
>
> We thank the reviewer for their thoughtful comments on our work. Below we respond to the questions raised.
>
> > The method is highly based on an existing method, YOTO. Although it doesn't need to train multiple models, the cost of YOTO is not discussed in this paper.
>
> We describe the computational cost of the YOTO model in extensive detail in Appendix E.1.1. including the computational resources used to train each YOTO model, the time required to train each YOTO model for each dataset, the model architecture for each dataset as well as the training details for the YOTO model. At a high-level, training one YOTO model takes roughly the same amount of time as taken by each of the separately trained models in our experiments. More specifically, on Adult and COMPAS datasets, training one YOTO model on a Tesla-V100-SXM2-32GB GPU takes roughly 5 minutes, whereas on the CelebA dataset training one YOTO model takes roughly 1.5 hours and on the Jigsaw dataset training one YOTO model takes roughly 6 hours.
>
> > The accuracy-fairness curve is algorithm-agnostic. Figure 4 shows that the estimated curve has more errors when using some particular algorithmic fairness methods. However, people tend to use some algorithmic fairness methods to train the model. In that case, the practical use of the curve is concerned.
>
>  We agree with the reviewer that the accuracy-fairness curve is algorithm-agnostic, and since our intervals are designed to align with the optimal trade-off curve, they should serve as reference for any model belonging to the model class $\mathcal{H}$ regardless of the algorithmic fairness method used to train the model. Note that in practice, any methodology whose trade-off lies above our upper confidence interval is likely to have a suboptimal trade-off, suggesting that alternative approaches may offer improved trade-offs.
>
> > What is the L_fair in equation (2)?
>
> We provided all of the smooth relaxations $L_{\text{fair}}$ considered in our work in Appendix E (pages 23 and 25). In our experiments, we considered a number of different fairness relaxations $L_{\text{fair}}$ as considered in previous fairness literature [Bendekgey & Sudderth, 2021]. For example, given a model $h_\theta : \mathcal{X} \rightarrow \mathbb{R}$, the relaxations for demographic parity considered in our baselines are of the form:
>  $$L_{\text{fair}}(h_\theta) = |\mathbb{E}[g(h_\theta(X))\mid A = 1] - \mathbb{E}[g(h_\theta(X))\mid A = 0]|$$
> Where the functions $g:\mathbb{R}\rightarrow \mathbb{R}$ considered in our experiments include $g(x) = x$ (denoted as 'linear' in our results), $g(x)=\log{\sigma(x)}$ where $\sigma(x):= 1/(1+\exp{-x})$ (used for the 'separate' baseline as well as YOTO models) and $g(x)=\log{\sigma(x)}$  (denoted as 'logsig' in our results). We will also include these details in the main text in the updated version of our paper.
>
> We hope our clarifications have addressed the reviewer's concerns, and kindly ask them to consider increasing their score.
>
> [Bendekgey & Sudderth, 2021] Harry Bendekgey and Erik B. Sudderth. Scalable and stable surrogates for flexible classifiers with fairness constraints.

---

### Official Review · Reviewer_H5sg · 2023-11-06

**Soundness:** 2 fair
**Presentation:** 2 fair
**Contribution:** 1 poor
**Rating:** 3
**Confidence:** 4

**Summary:**

This paper proposes a methodology to estimate the optimal fairness-accuracy trade-off curve for a given dataset and model class. The key ideas of the proposed method are two steps::


1. Use the You-Only-Train-Once (YOTO) framework to estimate the trade-off curve by training a single model.
2. Use the YOTO result to construct confidence intervals around the estimated trade-off curve using a held-out calibration dataset.


The claimed contributions of this paper are:
1. develop a method to calculate a range of allowed fairness violations for a given dataset and desired model accuracy.
2. Construct confidence intervals that statistically guarantee the optimal fairness-accuracy tradeoff curve.
3. Test the proposed method when sensitive attributes are scarce in the data.
4. Test the proposed method on different types of data. The intervals contained the tradeoff curves from state-of-the-art fairness methods like regularization and adversarial learning.

**Strengths:**

1. The confidence intervals of fairness seem needed in the fairness domain., however, it has problems.
2. The paper is clearly written and easy to follow. The graphics effectively illustrate the key ideas.

**Weaknesses:**

1. The statement "For a given dataset, model class, and accuracy, the permissible range of fairness violation is x to y." in this paper is problematic. Accuracy and fairness have an inherent connection (they exhibit trade-offs) and will influence each other. So accuracy cannot be the condition for the range of fairness violation.
2. How to evaluate whether the confidence interval is rational or correct? This does not seem to have ground truth for this and only visually evaluating figure 4 is not enough. There is a strong assumption that the curve learned with YOTO is the ground truth, this is not reasonable and even wrong. This strong but possibly wrong assumption is only mentioned in "In other words, under the assumption of large enough model capacity, training the loss-conditional YOTO model performs as well as the separately trained models while only requiring a single model." (Page 4). I strongly recommend treating this seriously.
3. The changeable fairness-accuracy trade-offs using one model may incur ethical issues, such as generating biased outcomes for certain groups of people. Based on this, I think this paper needs further ethical review.
4. The $\mathcal{L}_{fair}$ in Eq (2) is not presented at all. This paper should present the smooth relaxation of demographic parity.
5. What is the meaning of "Dataset Fairness" in the title? It seems this title is not suitable for the proposed method.
6. The experimental evaluation is not convincing to me. Since the Adult data is super imbalanced and the COMPAS data is small. I would suggest adding more experiments to really evaluate the proposed method, such as on the folktable dataset at https://github.com/socialfoundations/folktables


This paper does not meet the standards for acceptance to ICLR in its current form. For now, I would recommend rejection.

**Questions:**

Please address my concerns in the Weakness part.

----
----**After rebuttal**---

I thank the authors for their response and am sorry for the late response.  The authors' response addresses part of my concerns. But
1. The new explanation that "the lower confidence intervals presented in Proposition 3.4 depend on the gap between the YOTO achieved trade-off curve and the ground-truth trade-off curve" is essentially making a strong assumption that the YOTO should be good enough, although not group truth. I do not think this point is reasonable and grounded.
2. The evaluation based on such small tabular data is not convincing to me, without new results presented.

Based on the above and my original comments, I would maintain my original score.

**Details Of Ethics Concerns:**

The changeable fairness-accuracy trade-offs using one model may incur ethical issues, such as generating biased outcomes for certain groups of people. Based on this, I think this paper needs further ethical review.

---

> ### Author Response · Authors · 2023-11-17
> **Rebuttal by Authors - Part I**
>
> We thank the reviewer for providing comments on our work. Below we address the questions raised and try to clarify some of the misunderstandings:
>
> > The statement "For a given dataset, model class, and accuracy, the permissible range of fairness violation is x to y." in this paper is problematic. Accuracy and fairness have an inherent connection (they exhibit trade-offs) and will influence each other. So accuracy cannot be the condition for the range of fairness violation.
>
> We consider model fairness conditional on model accuracy **precisely because** of the accuracy-fairness trade-off. This is also in line with the fairness-constrained accuracy maximisation formulation provided in the fairness literature [Equation (3) in Agarwal et al, 2018]. Since accuracy and fairness can be at odds with each other and exhibit trade-offs, we should not consider the fairness violation value of a model in isolation.   As an example, if we do not constrain the model accuracy,  the minimum demographic parity for any given model class $\mathcal{H}$ will be 0 and will be attained by a model which makes a constant prediction regardless of the input. In this case, even though the minimum attainable demographic parity is 0, the model accuracy will likely be very low. Therefore, the goal when obtaining the optimal accuracy-fairness trade-off is to minimise the model fairness violation while constraining the model accuracy. We will further clarify this in the updated version of the paper.
>
> > How to evaluate whether the confidence interval is rational or correct? This does not seem to have ground truth for this and only visually evaluating figure 4 is not enough.
>
> Recall that the ground truth trade-off curve is defined as:
>
> $\tau^\ast_{\text{fair}}(\psi) \coloneqq \min_{h\in \mathcal{H}} \Phi_{\text{fair}}(h)$ subject to $\text{acc} \geq \psi$.
>
> For real-world datasets, this ground truth trade-off curve is intractable. This is because, in general, the constrained optimisation problem above is non-trivial to solve exactly. Furthermore, we only have access to a finite dataset making it impossible to obtain the exact trade-off curve in general. As a result of this, previous works in fairness literature only consider empirical accuracy-fairness trade-offs [Agarwal et al., 2018, Bendekgey & Sudderth, 2021, Zafar et al., 2015; 2017; 2019]. In this work, we construct the confidence intervals in an attempt to account for the approximation errors, and since the ground truth trade-offs are unknown, we empirically verify that the state-of-the-art methodologies lead to empirical trade-offs which are largely consistent with the obtained confidence intervals.
>
> Apart from this, we have conducted additional experiments with synthetic data setups where the ground truth is known. The experimental results included in the Appendix E.5 of the updated manuscript show that the empirical trade-off achieved by YOTO aligns very well with the ground truth trade-off curve. Additionally, the ground truth trade-off curve is covered by all the confidence intervals constructed.
>
> > There is a strong assumption that the curve learned with YOTO is the ground truth, this is not reasonable and even wrong.
>
> We wish to clarify that our work **does not** make the assumption that the curve learned with YOTO is the ground truth, and every theoretical result presented in our paper remains valid regardless of whether the YOTO trade-off curve is optimal or not. More specifically, as we mention in Section 3.2.1, the upper confidence intervals on $\tau^\ast_{\text{fair}}$ do not depend on how close the YOTO trade-off curve is to the ground truth optimum curve. In contrast, the lower confidence intervals presented in Proposition 3.4 depend on the gap between the YOTO achieved trade-off curve and the ground-truth trade-off curve (denoted by $\Delta(h)$). While this gap, $\Delta(h)$, is an unknown quantity in general, we propose sensitivity analysis techniques to posit plausible values for $\Delta(h)$ in practice in Appendix C.
>
> > This strong but possibly wrong assumption is only mentioned in "In other words, under the assumption of large enough model capacity, training the loss-conditional YOTO model performs as well as the separately trained models while only requiring a single model." (Page 4). I strongly recommend treating this seriously.
>
> The statement above does not refer to the optimality of the YOTO trade-off. Instead, the statement provides an intuitive, informal explanation of (Dosovitskiy & Djolonga, 2020, Proposition 1) and implies that under large enough model capacity, the trade-off curves obtained using YOTO should align with the trade-off curves obtained by training models separately (which may or may not be the same as optimal ground-truth trade-off curve). We will further clarify this in the updated version of our paper, by possibly including the formal result provided in (Dosovitskiy & Djolonga, 2020, Proposition 1) for completeness.

---

> > ### Author Response · Authors · 2023-11-17
> > **Rebuttal by Authors - Part II**
> >
> > > The changeable fairness-accuracy trade-offs using one model may incur ethical issues, such as generating biased outcomes for certain groups of people. Based on this, I think this paper needs further ethical review.
> >
> > Firstly, we note that the YOTO model is trained using fairness regularized loss, which is meant to ensure that model biases across different sensitive attributes are minimised. Moreover, [Dosovitskiy & Djolonga 2020, Proposition 1] proves that given a large enough model capacity, the YOTO model for a specific value of $\lambda'$ should be consistent with a separately trained model, trained with $\lambda'$ as the regularization parameter. This is also consistent with our empirical results in Figure 3 where the trade-off curve obtained using YOTO aligns very well with those obtained using separately trained models. Therefore, in terms of model fairness, there is little difference between using one YOTO model vs multiple separately trained models (although of course, YOTO is considerably more computationally efficient). In fact, we do not see any reason (either empirically or theoretically) that the YOTO model should be any less fair than the separately trained models.
> >
> > > The $\mathcal{L}_{\text{fair}}$ in Eq (2) is not presented at all. This paper should present the smooth relaxation of demographic parity.
> >
> > We provided all of the smooth relaxations $L_{\text{fair}}$ considered in our work in Appendix E (pages 23 and 25). In our experiments, we considered a number of different fairness relaxations $L_{\text{fair}}$ as considered in previous fairness literature [Bendekgey & Sudderth, 2021]. For example, given a model $h_{\theta} : \mathcal{X} \rightarrow \mathbb{R}$, the relaxations for demographic parity considered in our baselines are of the form:
> >
> > $$L_{\text{fair}}(h_\theta) = |\mathbb{E}[g(h_\theta(X))\mid A = 1] - \mathbb{E}[g(h_\theta(X))\mid A = 0]|$$
> >
> > Where the functions $g:\mathbb{R}\rightarrow \mathbb{R}$ considered in our experiments include $g(x) = x$ (denoted as 'linear' in our results), $g(x)=\log{\sigma(x)}$ where $\sigma(x):= 1/(1+\exp{-x})$ (used for the 'separate' baseline as well as YOTO models) and $g(x)=\log{\sigma(x)}$  (denoted as 'logsig' in our results). We will also include these details in the main text in the updated version of our paper.
> >
> > > What is the meaning of "Dataset Fairness" in the title? It seems this title is not suitable for the proposed method.
> >
> > The term "Dataset Fairness" is aimed to reflect that in this work we provide a methodology for quantifying the plausible range of accuracy-fairness trade-offs for any given dataset in a computationally efficient manner. This is also consistent with our argument that this trade-off can differ significantly across datasets, depending on factors such as dataset biases, and imbalances and therefore, applying fairness guidelines to datasets necessitates careful consideration to account for their distinct characteristics and underlying biases. To this end, our methodology provides dataset-specific insights and provides practitioners with a convenient tool for fairness decision-making tailored to the datasets under consideration. This is in contrast to previous works in fairness literature [Agarwal et al, 2018] which impose restrictions on model fairness apriori at training time, without taking into account the complete trade-off curve. We will clarify this further in the updated version of our paper.
> >
> > > The experimental evaluation is not convincing to me. Since the Adult data is super imbalanced and the COMPAS data is small. I would suggest adding more experiments ...
> >
> > We thank the reviewer for their suggestion. We would like to emphasise that in addition to the Adult and COMPAS datasets, we also consider the CelebA (202,599 datapoints)  and Jigsaw datasets (107,106 datapoints after pre-processing), both of which are significantly larger than Adult and COMPAS datasets. Moreover, these datasets are also comparatively more balanced with male/female split being 42%/58% in CelebA dataset and 47%/53% in Jigsaw dataset. Therefore, our experimental results demonstrate the effectiveness of our approach on datasets with varying sizes and imbalances. Additionally, training fair classifiers is likely to be more challenging on datasets with larger imbalances and smaller dataset sizes, since in these cases any reduction in fairness violation may come at the cost of a large reduction in model accuracy. Therefore, our experiments on Adult and COMPAS datasets show that our methodology remains effective even in these settings where training fair classifiers may be more challenging. Moreover, we will additionally include experimental evaluation on the folktable dataset in the updated version of our work.
> >
> > We hope that we were able to address all the reviewer's questions in the above and hope that the reviewer would consider increasing their score.

---

> > > ### Author Response · Authors · 2023-11-17
> > > **Rebuttal - References**
> > >
> > > [Agarwal et al. 2018] Alekh Agarwal, Alina Beygelzimer, Miroslav Dud´ık, John Langford, and Hanna Wallach. A reductions approach to fair classification. 03 2018.
> > >
> > > [Bendekgey & Sudderth, 2021] Harry Bendekgey and Erik B. Sudderth. Scalable and stable surrogates for flexible classifiers with fairness constraints.
> > >
> > > [Zafar et al. 2015] Muhammad Zafar, Isabel Valera, Manuel Rodriguez, and Krishna P. Gummadi. Fairness constraints: A mechanism for fair classification. 07 2015.
> > >
> > > [Zafar et al. 2017] Muhammad Bilal Zafar, Isabel Valera, Manuel Gomez Rodriguez, and Krishna P. Gummadi. Fairness beyond disparate treatment & disparate impact: Learning classification without disparate mistreatment.
> > >
> > > [Zafar et al. 2019] Muhammad Bilal Zafar, Isabel Valera, Manuel Gomez-Rodriguez, and Krishna P. Gummadi. Fairness constraints: A flexible approach for fair classification.
> > >
> > > [Dosovitskiy & Djolonga 2020] Alexey Dosovitskiy and Josip Djolonga. You only train once: Loss-conditional training of deep networks.

---

> ### Comment · Reviewer_H5sg · 2023-11-27
> **Thanks to Authors' Response**
>
> I thank the authors for their response and am sorry for the late response.  The authors' response addresses part of my concerns. But
> 1. The new explanation that "the lower confidence intervals presented in Proposition 3.4 depend on the gap between the YOTO achieved trade-off curve and the ground-truth trade-off curve" is essentially making a strong assumption that the YOTO should be good enough, although not group truth. I do not think this point is reasonable and grounded.
> 2. The evaluation based on such small tabular data is not convincing to me, without new results presented.
>
> Based on the above and my original comments, I would maintain my original score.
>
> (I also added the above comments in the original review.)

---

### Official Review · Reviewer_SKWL · 2023-11-06

**Soundness:** 1 poor
**Presentation:** 3 good
**Contribution:** 2 fair
**Rating:** 3
**Confidence:** 3

**Summary:**

Fairness-accuracy trade-off widely exists in machine learning models and fundamentally depends on dataset characteristics. Such dataset-dependent property impedes chasing universal fairness requirement across datasets. To this end, this paper proposes a computationally efficient approach to approximate the trade-off curve with statistical guarantees via adopting YOTO framework. The empirical results provide the guidelines in accuracy-constrained fairness decisions for various data modalities.

**Strengths:**

1.	The research problem on the fairness-accuracy trade-off is fundamental and important in machine learning fairness community.
2.	This paper is overall well-written and easy to follow.
3.	Due the unknown Pareto frontier of trade-off, the investigation of trade-off with confidence interval with statistical guarantee makes much sense to me.

**Weaknesses:**

1.	Motivation. Can the authors elaborate on the motivation for using a universe fairness requirement across datasets? What are the advantages of doing this?
2.	Technique novelty. This paper introduces a computationally efficient method for estimating the trade-off. However, from my understanding, the efficiency part directly adopts YOTO framework and the confidence interval estimation only involves trivial bounds.
3.	Pareto frontier. The achievable trade-off by YOTO may not be consistent with the ground-truth Pareto optimum. It seems that this paper is over-claimed since the true Pareto trade-off investigation is not touched. Additionally, how do you use a universe fairness requirement across datasets? The approximated trade-off seems not be a good choice since the gap between achievable trade-off by YOTO and ground-truth Pareto optimum may also be dataset-dependent.
4.	Experiments. (a) The evaluation of the confidence interval is vague. It seems that the conservative estimation is never penalized by the current results, such as Figures 3 and 4. Which confidence interval estimation method is better? (b) In Section 3.1, the author mentioned $\lambda$ in Eq. (2) offers litter control over the accuracy, which is counter-intuitive for such regularization. Can you provide experimental results to further support this statement? (c) Is it possible to create a synthetic dataset with a known ground-truth trade-off in the experiments? Otherwise, many conclusions can only hold for the achievable trade-off by YOTO. (d) There are confidence intervals for both accuracy and fairness. How can you plot these two intervals in Figures 3 and 4?
5.	From my understanding, the optimization for fairness with accuracy and the optimization for accuracy with fairness constraint have the same trade-off. I am curiosu why the authors select the former one and highlight the difference in the first paragraph of section 2.1.

**Questions:**

Please see weakness part.

---

> ### Author Response · Authors · 2023-11-17
> **Rebuttal by Authors - Part I**
>
> We would like to thank the reviewer for taking the time to review our work. There seem to be some misunderstandings regarding our methodology which we seek to clarify below.
>
> > Motivation. Can the authors elaborate on the motivation for using a universe fairness requirement across datasets? What are the advantages of doing this?
>
> We would like to emphasise that, in this work, we argue **against** the use of a universal fairness requirement across datasets because the accuracy-fairness trade-off is highly dependent on the characteristics of the dataset.  Therefore, previous works such as [Agarwal et al., 2018 ] which use a uniform fairness requirement (such as requiring the Demographic Parity to be $\leq 0.1$ regardless of the dataset) may have varying implications for the achievable model accuracy across the different datasets. Our proposed methodology instead allows practitioners to quantify the complete accuracy-fairness tradeoff curve which may be used to obtain the range of permissible fairness violations at any given level of model accuracy chosen at inference time and does not require imposing any restrictions on model fairness or accuracy apriori at training time. We will further clarify this distinction in the next version of our paper.
>
> > Technique novelty. This paper introduces a computationally efficient method for estimating the trade-off. However, from my understanding, the efficiency part directly adopts YOTO framework and the confidence interval estimation only involves trivial bounds.
>
> There are a number of important technically novel aspects regarding both, the construction of confidence intervals and the application of the YOTO framework in our methodology, which we believe will be of interest to the ML fairness community:
> 1. Firstly, we would like to emphasise that our method of constructing the confidence intervals on the trade-off curve $\tau^\ast_{\text{fair}}$ does not simply rely on trivial bounds. While standard methods like Hoeffding's inequality can be used directly to obtain confidence intervals on model accuracy, the same is not true for fairness violation metrics like Demographic Parity, Equalised Odds and Equalized Opportunity which involve multiple (conditional) expectation terms. While a naive application of union bounds can be used to construct confidence intervals on these fairness violation metrics, these bounds are known to be highly conservative and are often non-informative. Therefore, in order to construct informative confidence intervals on fairness violations, we propose a subsampling procedure in Appendix B.2 which provides a novel methodology of constructing valid confidence intervals on fairness violation metrics by subsampling calibration data. We will include a discussion of this approach in the main text of our paper.
> 2.  Secondly, our confidence intervals not only incorporate finite sample uncertainty from finite calibration data, but also account for the possible sub-optimality in the fairness trade-offs achieved by $h$. Specifically, to incorporate the latter, we formally consider the gap between the optimal accuracy-fairness trade-off $\tau^\ast_{\text{fair}}$ and the trade-off achieved by a model $h$, denoted by $\Delta(h)$ in Figure 2 in the main text. Part of our methodology for obtaining the lower confidence interval involves quantifying this gap $\Delta(h)$. While this gap $\Delta(h)$ is intractable in general, we propose a sensitivity analysis methodology to posit plausible value for $\Delta(h)$ in practice (in Appendix C). In contrast to the previous works in fairness literature [Agarwal et al., 2018, Bendekgey & Sudderth, 2021, Zafar et al., 2015; 2017; 2019], to the best of our knowledge, our work is the first to consider uncertainty arising from both, finite datasets used to evaluate accuracy and fairness metrics as well as the possible sub-optimality in the fairness trade-offs achieved by $h$.
> 3. Thirdly, we also propose a novel methodology for constructing informative confidence intervals in the setting where the sensitive attributes $A$ are accessible only for a small subset of the calibration dataset. This methodology effectively combines data with actual and predicted sensitive attributes to derive tighter and more accurate CIs, even when the majority of $A$ values are absent.
> 4. Finally, while YOTO has previously been applied in areas like multi-task learning [Lin et al. 2020], our work provides a novel application of the YOTO framework to fairness literature and offers practitioners the capability to, at inference time, specify desired accuracy levels and promptly receive corresponding admissible fairness violation ranges.

---

> > ### Author Response · Authors · 2023-11-17
> > **Rebuttal by Authors - Part II**
> >
> > > Pareto frontier. The achievable trade-off by YOTO may not be consistent with the ground-truth Pareto optimum. It seems that this paper is over-claimed since the true Pareto trade-off investigation is not touched.
> >
> > We do, in-fact, consider the gap between YOTO achievable trade-off and ground-truth Pareto optimum when constructing the confidence intervals. Specifically, the $\Delta(h)$ term in the lower confidence intervals (Proposition 3.4) quantifies this gap (Figure 2 also provides a visual representation of this gap). Moreover, the upper confidence intervals obtained in Proposition 3.2 do not depend on the gap between YOTO achievable trade-off and ground-truth Pareto optimum and remain valid even if the YOTO model achieves sub-optimal accuracy-fairness trade-offs. On the other hand, the lower confidence intervals obtained in Proposition 3.4 depend on the  $\Delta(h)$ term which is intractable in general. To remediate this, we employ sensitivity analysis techniques to incorporate any belief on plausible values for $\Delta(h)$ in Appendix C. We will further clarify this in the updated version of our paper.
> >
> > > Additionally, how do you use a universe fairness requirement across datasets? The approximated trade-off seems not be a good choice since the gap between achievable trade-off by YOTO and ground-truth Pareto optimum may also be dataset-dependent.
> >
> > As we emphasise in our response above, in our work we argue **against** the use of a universal requirement across different datasets. It is true that the gap between YOTO achievable trade-offs and the ground-truth Pareto optimum may be data dependent. Our confidence intervals in Proposition 3.4 take this gap into account explicitly using the $\Delta(h)$ term. In this case, the use of sensitivity analysis described in Appendix C may be used to posit plausible values for this gap in practice using the calibration data.
> >
> > > The evaluation of the confidence interval is vague. It seems that the conservative estimation is never penalized by the current results, such as Figures 3 and 4. Which confidence interval estimation method is better?
> >
> > There is an inherent trade-off between informativeness and validity of confidence intervals. For example, while the intervals obtained using Hoeffding's inequality satisfy finite sample coverage guarantees, they are conservative as can be seen in Figures 3 and 4 of our experiments. Therefore, the range of plausible fairness values for any given accuracy threshold may be very wide and not very informative when using these intervals. In contrast, bootstrap confidence intervals are comparatively tighter and therefore lead to a more informative range of plausible fairness values, although they come at the cost of only asymptotic coverage guarantees. Which of these confidence intervals to use in practice will depend on the validity guarantees sought by the practitioners.
> >
> > Furthermore, we would like to note that our intervals are designed to align with the optimal trade-off curve and therefore, any methodology whose trade-off lies above our upper bound is likely suboptimal, suggesting that alternative approaches may offer improved trade-offs.
> >
> > > In Section 3.1, the author mentioned $\lambda$ in Eq. (2) offers little control over accuracy, which is counter-intuitive for such regularization. Can you provide experimental results to further support this statement?
> >
> > Our experimental results provided in Figure 3 demonstrate this phenomenon for regularizations based baselines such as 'separate', 'logsig' and 'linear'. Specifically, these results are obtained by training model using regularized losses, with regularization parameters $\lambda$ chosen over a fine grid (of size 100) for $0\leq \lambda \leq 5$. However, the accuracy-fairness trade-offs obtained exhibit high variance in many cases (see Jigsaw results in Figure 3, for example). This shows that changing $\lambda$ values only slightly can lead to a large shift in the empirical accuracy-fairness trade-offs obtained for the models trained with these losses. This is also consistent with the findings in prior works (Bendekgey & Sudderth, 2021), where the trade-off curves obtained exhibit high variance and do not vary smoothly with varying $\lambda$.

---

> > > ### Author Response · Authors · 2023-11-17
> > > **Rebuttal by Authors - Part III**
> > >
> > > > Is it possible to create a synthetic dataset with a known ground-truth trade-off in the experiments? Otherwise, many conclusions can only hold for the achievable trade-off by YOTO.
> > >
> > > We thank the reviewer for their suggestion. We have conducted additional experiments with a synthetic dataset with a known ground-truth trade-off. The results which are provided in Appendix E.5 of our updated manuscript show that the empirical trade-off achieved by YOTO aligns very well with the ground truth trade-off curve. Additionally, the ground truth trade-off curve is covered by all the confidence intervals constructed. This simple setting serves as a sanity check to make sure that the trade-offs obtained using YOTO are consistent with the ground truth and that the intervals obtained using our methodology are well-calibrated.
> > >
> > > >There are confidence intervals for both accuracy and fairness. How can you plot these two intervals in Figures 3 and 4?
> > >
> > > As we explain in Propositions 3.2 and 3.4, the confidence intervals for accuracy and fairness can be used to obtain confidence on the optimum achievable trade-off $\tau^\ast_{\text{fair}}$ of the form:
> > > $$\mathbb{P}(\tau^\ast_{\text{fair}}(\Psi) \in \Gamma_{\text{fair}}^\ast) \geq 1-\alpha$$
> > > In Figures 3 and 4, we do not plot the confidence intervals for accuracy and fairness individually, but instead, we use the results in Propositions 3.2 and 3.4 to plot the two-sided confidence intervals $\Gamma^\ast_{\text{fair}}$ on optimum trade-off $\tau^\ast_{\text{fair}}$ directly.
> > >
> > > Intuitively, to understand how these confidence intervals are obtained, consider the example where the accuracy of a model $h$ is  $\geq 90\\%$, and the DP of $h$ is  $\leq 15\\%$. Then the worst-case accuracy-fairness trade-off of model $h$ will be achieved when $\text{acc}(h) = 90\\%$ and $\Phi_{\text{DP}}(h)=15\\%$ (i.e., accuracy is minimised and the demographic parity is maximised). Since $\tau^\ast_{\text{fair}}$ denotes the ground-truth optimum trade-off, it must not achieve a worse accuracy-fairness trade-off than the model $h$, i.e.  $\tau^\ast_{\text{fair}}(\text{acc}(h)) \leq \Phi_{\text{DP}}(h)$ and therefore, in the worst-case, $\tau^\ast_{\text{fair}}(90\\%) \leq 15\\%$. This provides us with a valid upper bound on the accuracy-fairness trade-off curve at accuracy = 90%. Next, to obtain an upper confidence interval on each accuracy value, we consider different values of $\lambda$ in our YOTO model, each of which leads to a different worst-case trade-off and provides an upper bound for the curve at different accuracy values. We will further make this clearer in the next version of our paper. The lower bound can be analogously determined, although this involves additional consideration as outlined in Section 3.2.2.
> > >
> > > > From my understanding, the optimization for fairness with accuracy and the optimization for accuracy with fairness constraint have the same trade-off. I am curious why the authors select the former one and highlight the difference in the first paragraph of section 2.1.
> > >
> > > It is true that optimization for fairness with accuracy constraint and the optimization for accuracy with fairness constraint should lead to the same trade-off. However, in our problem setup in Section 2.1, we specifically choose the former formulation so as not to impose constraints on fairness violations apriori. This is in line with our argument that setting a uniform requirement for fairness regardless of the dataset ignores the dataset-specific accuracy-fairness trade-offs. To this end, our methodology approximates the complete trade-off curve $\tau^\ast_{\text{fair}}$ without imposing apriori restrictions on either model fairness or model accuracy at training time unlike [Agarwal et al, 2018]. We will further clarify this in the updated version of our paper.
> > >
> > > We hope that the above has addressed the questions raised by the reviewer adequately, and that the reviewer will consider raising their score.
> > >
> > > [Agarwal et al. 2018] Alekh Agarwal, Alina Beygelzimer, Miroslav Dud´ık, John Langford, and Hanna Wallach. A reductions approach to fair classification. 03 2018.
> > >
> > > [Bendekgey & Sudderth, 2021] Harry Bendekgey and Erik B. Sudderth. Scalable and stable surrogates for flexible classifiers with fairness constraints.
> > >
> > > [Zafar et al. 2015] Muhammad Zafar, Isabel Valera, Manuel Rodriguez, and Krishna P. Gummadi. Fairness constraints: A mechanism for fair classification. 07 2015.
> > >
> > > [Zafar et al. 2017] Muhammad Bilal Zafar, Isabel Valera, Manuel Gomez Rodriguez, and Krishna P. Gummadi. Fairness beyond disparate treatment & disparate impact: Learning classification without disparate mistreatment.
> > >
> > > [Zafar et al. 2019] Muhammad Bilal Zafar, Isabel Valera, Manuel Gomez-Rodriguez, and Krishna P. Gummadi. Fairness constraints: A flexible approach for fair classification.
> > >
> > > [Lin et al. 2020] Xi Lin, Zhiyuan Yang, Qingfu Zhang, and Sam Tak Wu Kwong. Controllable pareto multi-task learning.

---

### Author Response · Authors · 2023-11-17
**Author Rebuttal -- Additional experiments**

Firstly, we would like to thank the reviewers for taking the time to review our paper and providing many insightful comments regarding it. To address some of the concerns raised, we have conducted additional experiments with a synthetic data setting (where the ground truth trade-off can be obtained) and included the results in Appendix E.5 of our updated manuscript (page 30). This is the *only* change in our updated manuscript. We hope that our rebuttal, along with the additional results, answers the reviewers' questions, and that the reviewers will consider increasing their scores.

---

### Meta-Review · Area_Chair_49qu · 2023-12-13

**Metareview:**

This paper applies the YOTO framework to capture the accuracy-fairness trade-off of a given model class trained on a given data set. The authors use conformal prediction to obtain confidence intervals for the Pareto-front. While this is a novel application of YOTO and the paper is relatively well written, many technical details are omitted or not well referenced in the main body of the paper. Reviewers were confused and lacked critical technical details, a.o. because the authors did not relate to the relevant Appendices or justified their claims.

After a careful consideration of the authors concerns that reviewers missed key aspects of their work, I have to conclude that the main body of the paper is not a stand-alone write-up. For instance, the motivation/justification for adopting a different constraint optimisation problem than is typically done in the literature is not provided; the methodology for handling sensitive attributes lacks technical details; the experimental set-up described in the paper is does not mention the model class considered; the smooth relaxation used in Eq. (2) had to clarified during the rebuttal; etc.

This paper proposes some interesting directions and seems to be making valuable contributions. Unfortunately, the authors should do a better job at removing ambiguities and articulating their contributions, their experimental validation, and the gains their method provides in practice. While I understand not all the feedback provide by the reviewers was constructive, it provides indications how this work and its presentation can be improved.

**Justification For Why Not Higher Score:**

While the narrative of the paper is clear, it is difficult to asses whether the claims are supported and what the gains provided by the proposed method are in practice. Reviewers were also confused about a number of contributions and technical details.

**Justification For Why Not Lower Score:**

N/A

---

### Decision · Program_Chairs · 2024-01-16

Reject